# 🔍 GLANCE AND FOCUS REINFORCEMENT FOR PAN-CANCER SCREENING

**Linshan Wu, Jiaxin Zhuang & Hao Chen**[*]
Department of Computer Science and Engineering
The Hong Kong University of Science and Technology
Hong Kong, China
{linshan.wu,jzhuangad}@connect.ust.hk, jhc@cse.ust.hk

## ABSTRACT

Pan-cancer screening in large-scale CT scans remains challenging for existing AI methods, primarily due to the difficulty of localizing diverse types of tiny lesions in large CT volumes. The extreme foreground-background imbalance significantly hinders models from focusing on diseased regions, while redundant focus on healthy regions not only decreases the efficiency but also increases false positives. Inspired by radiologists' glance and focus diagnostic strategy, we introduce **GF-Screen**, a Glance and Focus reinforcement learning framework for pan-cancer screening. GF-Screen employs a Glance model to localize the diseased regions and a Focus model to precisely segment the lesions, where segmentation results of the Focus model are leveraged to reward the Glance model via Reinforcement Learning (RL). Specifically, the Glance model crops a group of sub-volumes from the entire CT volume and learns to select the sub-volumes with lesions for the Focus model to segment. Given that the selecting operation is non-differentiable for segmentation training, we propose to employ the segmentation results to reward the Glance model. To optimize the Glance model, we introduce a novel group relative learning paradigm, which employs group relative comparison to prioritize high-advantage predictions and discard low-advantage predictions within sub-volume groups, not only improving efficiency but also reducing false positives. In this way, for the first time, we effectively extend cutting-edge RL techniques to tackle the specific challenges in pan-cancer screening. We conduct training and validation on a large-scale pan-cancer dataset comprising 5,117 CT scans. Extensive experiments on 16 internal and 7 external datasets across 9 lesion types demonstrated the effectiveness of GF-Screen. Notably, GF-Screen leads the public validation leaderboard of MICCAI FLARE25 pan-cancer challenge, surpassing the FLARE24 champion solution by a large margin (+25.6% DSC and +28.2% NSD). In addition, through discarding redundant regions, GF-Screen reduces the computation costs by 5.7 times, significantly improving inference efficiency. The superior performance of GF-Screen remarks a novel and practical breakthrough in pan-cancer screening. Code is available at https://github.com/Luffy03/GF-Screen.

## 1 INTRODUCTION

Cancer is a leading cause of death worldwide (Bray et al., 2024), and effective cancer screening is crucial to reducing the mortality rate of patients (Shieh et al., 2016; McKinney et al., 2020; Jiang et al., 2022). AI-driven cancer screening in large-scale Computed Tomography (CT) scans has received increasing attention in clinical applications (Isensee et al., 2021; Cao et al., 2023; Hu et al., 2025), since CT is a low-cost and commonly-used imaging protocol in routine physical examination (Pickhardt et al., 2020; 2023). Specifically, pan-cancer screening aims to develop one universal model to detect and segment different types of lesions in large-scale CT scans (Ma et al., 2024; Chen et al., 2023; Jiang et al., 2024; 2025), which has profound significance in clinical practice.

---

[*]Corresponding Author.

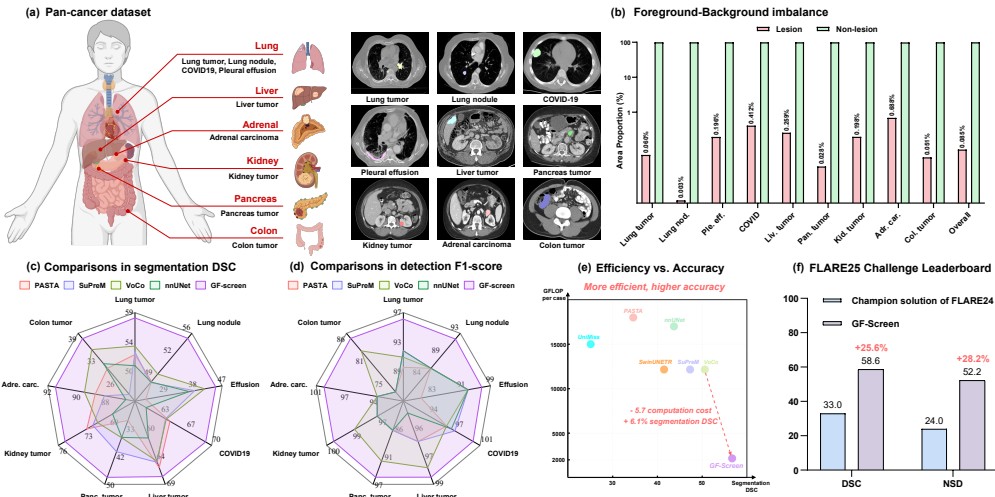

Figure 1: (a) The pan-cancer dataset used in this study, encompassing 5,117 CT scans across 9 different types of lesions from 16 internal and 7 external datasets. (b) Significant foreground-background imbalance: in our dataset, lesions occupy only 0.085% area proportions in CT volumes. (c) Comparisons in pan-cancer segmentation. (d) Comparisons in pan-cancer detection. (e) Inference efficiency (GFLOP per scan) and segmentation DSC on the FLARE23 validation dataset. Compared with the second-best model, GF-Screen is 5.7× faster with higher segmentation DSC. (f) Comparisons on the FLARE25 challenge validation leaderboard. GF-Screen outperforms the second-ranked algorithm (champion solution of FLARE24) by a large margin (+25.6% DSC and +28.2% NSD).

Although some promising results have been demonstrated in recent cancer screening works (Yan et al., 2018; Cao et al., 2023; Hu et al., 2025; Zhang et al., 2025a), it remains challenging for the development of pan-cancer screening models (Ma et al., 2024). Primarily, lesions typically occupy small areas in large CT volumes. The extreme foreground-background imbalance poses a significant obstacle for models to localize diverse lesion types across different organs. In addition, the redundant focus on healthy regions not only leads to higher false positives but also impedes the screening efficiency, presenting a critical barrier to real-world deployment of AI models (Ma et al., 2024). To address these challenges, we highlight that effective pan-cancer screening models should be capable of focusing on diseased regions while minimizing redundant focus on healthy regions.

In contrast to AI models, experienced radiologists can rapidly disregard irrelevant regions and concentrate their diagnostic attention on potentially diseased regions during cancer screening. Typically, radiologists will glance at the entire CT volume, then focus on specific regions for precise diagnosis (Wang et al., 2025a), which inspires us to explore a similar *glance and focus* strategy in AI models. In this work, we introduce **GF-Screen**, a Glance and Focus reinforcement learning framework for pan-cancer screening. GF-Screen adopts a Glance model to localize the diseased regions at a coarse level and a Focus model to precisely segment lesions at a finer level, with both models operating synergistically. Specifically, the Glance model crops a group of sub-volumes from the whole CT volume and learns to select those containing lesions for the Focus model to segment. Given that the selecting operation is non-differentiable for segmentation training, we propose to employ RL to optimize the Glance model by leveraging segmentation results from the Focus model. We carefully design a simple-yet-effective reward function for sub-volume selection, *i.e.*, assigns high advantages to sub-volumes where the Focus model can accurately segment lesions and low advantages otherwise. To optimize the Glance model, we introduce a novel group relative learning paradigm, which employs group relative comparison to prioritize high-advantage predictions and discard low-advantage predictions within sub-volume groups. In this way, we effectively encourage the model to focus on diseased regions and discard redundant healthy regions, which not only improves efficiency but also reduces false positives.

To evaluate the effectiveness, we aggregate 5,117 CT scans from 23 public datasets for training and validation. Extensive experiments on 16 internal and 7 external datasets across 9 different lesion types highlight the superiority of GF-Screen. By focusing on diseased regions and discarding redun-

dant healthy regions, GF-Screen reduces the computation costs by an average of 5.7 times. Notably, GF-Screen leads the competitive MICCAI FLARE25 pan-cancer challenge validation leaderboard, which is one of the most representative challenges in this field. On the public validation leaderboard, GF-Screen surpasses the second-ranked algorithm (champion solution of FLARE24) by +25.6% DSC and +28.2% NSD, marking a novel and practical breakthrough in pan-cancer screening.

## 2    RELATED WORKS

### 2.1    CANCER SCREENING

AI-driven cancer screening has witnessed rapid development in recent years, which plays an important role in improving patient survival rates (Cao et al., 2023; Hu et al., 2025). Existing works primarily rely on segmentation models (Isensee et al., 2021; Wu et al., 2025b; 2024), which segment lesions at the pixel level, precisely outlining the lesion sizes and positions for cancer screening. Despite their promising results, these approaches are often specialized to one single type of lesion. Moving forward, pan-cancer screening aims to employ one model for detecting diverse cancers, presenting greater challenges but promising potential for clinical practice (Chen et al., 2023; Ma et al., 2024; Jiang et al., 2024; Lei et al., 2025; He et al., 2024; Chen et al., 2025). Specifically, CancerUniT (Chen et al., 2023) employed a query-based Mask-Transformer (Cheng et al., 2022) to segment eight types of lesions. ZePT (Jiang et al., 2024) focused on the zero-shot segmentation of unseen lesion types. PASTA (Lei et al., 2025) pre-trained a pan-cancer foundation model via tumor synthesis and segmentation.

Although decent performance has been demonstrated, existing works predominantly overlook the critical foreground-background imbalance in cancer screening. Lesions typically occupy small areas in large CT volumes (Miller et al., 1981; Bassi et al., 2025), making it challenging for models to focus on tiny diseased regions. As shown in Fig. 2, previous cancer screening methods (Isensee et al., 2021; Cao et al., 2023; Hu et al., 2025; Liu et al., 2023; Tang et al., 2022; Chen et al., 2024a;b; Lei et al., 2025) generally employ slide-window inference on large CT volumes without discarding the redundant healthy regions, which not only impedes the inference efficiency but also increases false positives on healthy regions. To this end, we develop GF-Screen to address this significant challenge by mimicking the glance-and-focus diagnostic strategy of radiologists, aiming to focus on diseased regions while ignoring redundant healthy regions.

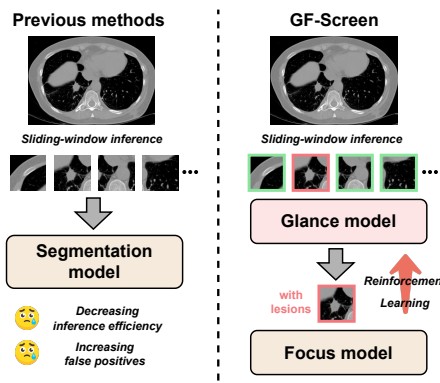

Figure 2: Comparison between previous cancer screening methods and GF-Screen.

### 2.2    REINFORCEMENT LEARNING IN VISION PERCEPTION

RL has demonstrated remarkable effectiveness in vision perception tasks. Recent advanced vision-language models (Shen et al., 2025; Yue et al., 2025; Wang et al., 2025b; Zhang et al., 2025b) explored the vision reasoning ability via RL. However, most of these works employ RL to optimize the large language models. In pure vision tasks, RL can serve as a tool for adaptive visual search (Wang et al., 2020; Huang et al., 2022; Wang et al., 2021; Pardyl et al., 2024). For example, Wang et al. (2020) proposed to train an actor-critic model (Haarnoja et al., 2018) via Proximal Policy Optimization (PPO) (Schulman et al., 2017) for discarding redundant patches in image classification, improving efficiency without a significant performance drop. However, it fails to tackle dense image parsing tasks such as detection and segmentation, and also requires training an extra value model for advantage estimation. In medical image analysis, Hao et al. (2022) proposed an RL method to imitate human visual search behavior for lesion localization in X-ray images. In this work, we extend the state-of-the-art RL technique to solve the specific challenges in pan-cancer screening, establishing a novel and practical pioneer for the RL applications in medical vision tasks.

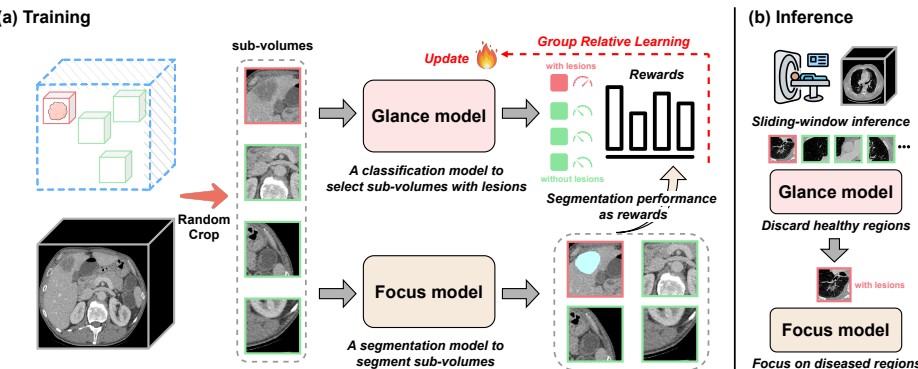

Figure 3: The overall framework of GF-Screen, including a Glance model to localize diseased regions and a Focus model to precisely segment lesions. (a) In the training stage, we conduct segmentation on all sub-volumes and leverage the segmentation results to reward the Glance model via a novel group-relative learning paradigm. (b) In the inference stage, a dynamic number of sub-volumes classified as "with lesions" by the Glance model will be input to the Focus model for segmentation, where the redundant regions will be discarded.

## 3 METHOD

### 3.1 TOWARDS PRECISE AND EFFICIENT PAN-CANCER SCREENING

In this work, we develop a simple-yet-effective framework, enabling precise and efficient pan-cancer screening. As illustrated in Fig. 3, GF-Screen comprises two key components: a Glance model for coarse-level localization and a Focus model for pixel-level segmentation. The Glance model is a lightweight classification model, which can efficiently categorize the cropped sub-volumes as either "with lesions" or "without lesions". The Focus model can deliver precise segmentation of lesions, providing important positions and sizes information for cancer screening. Both models allow seamless integration of advanced network architectures.

These two models operate synergistically during training and inference. During training, as shown in Fig. 3(a), randomly cropped sub-volumes are fed into both models. The Focus model is supervised by lesion masks in segmentation training, using a typical combination of per-pixel binary cross-entropy loss and dice loss following previous medical image segmentation methods. The Glance model is trained via RL, where rewards are derived from the Focus model's segmentation results. In the inference stage (Fig. 3(b)), the input CT volume is first divided into a group of sub-volumes using a sliding-window approach as in previous methods. The Glance model then filters out healthy sub-volumes, forwarding only lesion-containing sub-volumes to the Focus model for precise segmentation. In this way, GF-Screen improves inference efficiency by eliminating wasted computation on healthy regions, while also reducing false positives.

### 3.2 GLANCE AND FOCUS REINFORCEMENT LEARNING

In our framework, the most critical challenge is ***how to effectively train the Glance model for discarding redundant healthy regions without compromising lesion recognition performance***. Since we have lesion masks for supervision, a straightforward way is to degrade the lesion masks $m$ as binary categories $y$ ("with lesions" or "without lesions") for each sub-volume $v_i$, then employing a typical cross-entropy loss for classification training as follows:

$$\mathbb{CE}(o_i, y_i) = -y_i * log(o_i) - (1 - y_i) * (1 - log(o_i)), \quad o_i = G(v_i), \tag{1}$$

where $o_i$ denotes the selection outputs of the Glance model $G$. However, we observe that there are two fundamental shortcomings within this training approach:

- Lesions suffer from a severe foreground-background imbalance in CT volumes, meaning most sub-volumes contain no lesions. This class imbalance causes the Glance model to overfit on negative cases, lowering its sensitivity to positives. Consequently, it fails to effectively select and forward diseased regions to the Focus model.

- As shown in Fig. 4, due to random crop, many lesion-containing sub-volumes present challenges caused by either partial inclusion or suboptimal viewing angles. If we simply degrade the lesion masks as classification labels for training, the classification training of these sub-volumes would be significantly hampered.

Motivated by these challenges, we aim to develop a more effective training paradigm for the Glance model. As shown in Fig. 4, experienced radiologists typically adopt an intelligent viewing strategy, *i.e.*, they generally focus on the most diagnostically informative perspectives for accurate cancer diagnosis. GF-Screen emulates this clinical expertise in AI models, which is developed to not only select the sub-volumes with lesions, but also prioritize the optimal views within the sub-volume groups for more precise segmentation.

To this end, we propose to *leverage the segmentation results of the Focus model to supervise the Glance model*. However, it is worth noting that the selection operation of the Glance model is non-differentiable in the segmentation training of the Focus model. To address this challenge, we introduce RL techniques for training the Glance model. Specifically, in our RL framework, the Glance model acts as a policy model to deliver actions (select this sub-volume or not), while the Focus model serves as a reward model to reward the Glance model based on the segmentation results. In this way, we effectively enable the Glance model to discard redundant regions and prioritize the optimal diagnostic views that can yield better segmentation results.

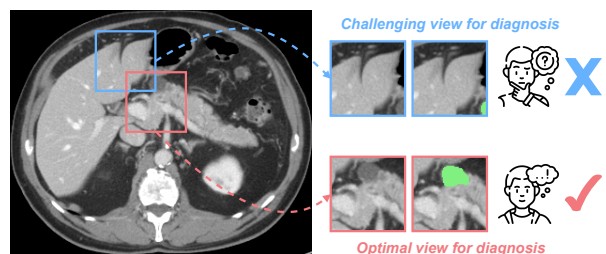

Figure 4: Illustration of sub-volume variation. The blue regions represent the challenging view with partial lesions and poor angles. While red regions indicate the optimal diagnostic view containing complete lesion information, generally with more precise segmentation results. Thus, we propose to leverage segmentation results as reward signals for RL.

### 3.3 GROUP RELATIVE LEARNING FOR SUB-VOLUMES SELECTION

First, the Glance model acts as a policy model to deliver actions. The final layer of the policy model is a 2-unit layer with softmax activation, outputting a probability distribution over the two actions (select (1) or discard (0)). During training, the output $o$ is stochastically sampled from this softmax distribution. As with standard RL algorithms, we can directly optimize the underlying probability distribution using the policy gradient, bypassing the non-differentiability of the sampling step.

**Reward design**. Our reward function is derived from the segmentation results of the Focus model, offering a simple-yet-effective mechanism. Given a group of $N$ cropped sub-volumes $\{v_1, v_2, ..., v_N\}$, the Focus model $F$ generate segmentation prediction $s_i$ for each sub-volume $v_i$. The reward $r_i$ is binary and determined by the overlap between the predicted segmentation $s_i$ and the ground-truth lesion mask $m_i$. Once the segmentation prediction $s_i$ overlaps with the lesion mask $m_i$, it returns rewards $r_i = 1$; otherwise, it returns $r_i = 0$:

$$r_i = \mathbb{1}(s_i \cap m_i \neq \emptyset), s_i = F(v_i). \tag{2}$$

We have further explored using the segmentation DSC for a more granular reward signal, but we observed that the performance is worse. We conclude that detection of the lesion presence is more important in the Glance model, while DSC varies significantly with lesion complexity (*e.g.*, types, sizes, and positions). Concretely, a high segmentation DSC may simply reflect an "easy" sub-volume (*e.g.*, one with clear boundaries or canonical orientation). Conversely, a low segmentation DSC might indicate a challenging but clinically critical case (*e.g.*, partial lesions or suboptimal viewing angles). Thus, to prioritize detection accuracy in the Glance model, we use the binary detection reward during RL training. Previous cancer screening works (Cao et al., 2023; Hu et al., 2025) also employed such detection methods to avoid discarding hard yet important cases that with lower segmentation scores, substantiating the reasonableness of our reward design.

**Group Relative Learning (GRL)**. Previous RL approaches for visual perception tasks (Wang et al., 2020; Huang et al., 2022) typically employ traditional policy optimization methods like PPO. How-

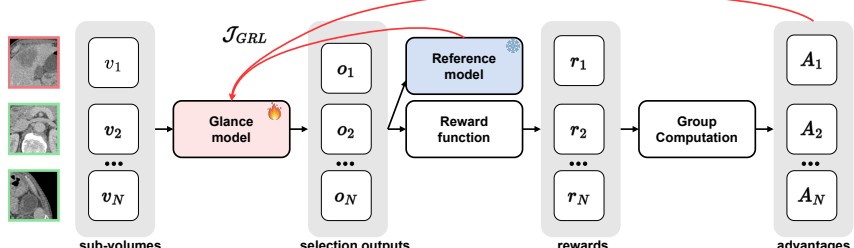

Figure 5: The Group Relative Learning (GRL) paradigm in GF-Screen. The Glance model $G$ is trainable while the reference model $G_{ref}$ is frozen. We first generate selection outputs $o$ from the input sub-volumes $v$, then use the reward function Eq. 2 to calculate the rewards $r$. Finally, we compute the relative advantages $A$ via the GAE function Eq. 3.

ever, PPO requires training a large value model as a critic to estimate the advantages, which brings a substantial memory and computational burden. In comparison, GRPO (Shao et al., 2024) foregoes the value model, instead using Large Language Models (LLMs) to generate a group of candidate answers and estimate the relative advantages, effectively obviating the need for training an extra value model and achieving better performance. GRPO (Shao et al., 2024) has demonstrated remarkable success in recent LLMs, where multiple candidate responses can be naturally generated. However, it remains challenging to transfer GRPO to vision tasks due to the difficulty in generating a group of candidate results without LLMs.

Our GF-Screen naturally overcomes this limitation: **the group of cropped sub-volumes readily provides comparable candidates**. This unique feature enables us to conduct reward comparison across sub-volumes for relative advantage estimation, eliminating the requirement of additional candidate generation mechanisms and achieving seamless integration of group relative learning in GF-Screen. Moreover, the group relative advantage comparison effectively enables us to *prioritize high-advantage predictions and discard low-advantage predictions* within sub-volume groups. Without the usage of LLMs, GF-Screen effectively shifts the paradigm of GRPO from NLP to vision tasks.

The group relative learning paradigm in GF-Screen is shown in Fig. 5. We first calculate the relative advantages of sub-volume groups by Generalized Advantage Estimation (GAE). Specifically, we employ group computation to normalize the rewards by computing the mean and standard deviation (std) of rewards, subsequently deriving the advantage as:

$$A_i = \frac{r_i - mean\{r_1, r_2, ..., r_N\}}{std\{r_1, r_2, ..., r_N\}}, \tag{3}$$

where $A_i$ represents the advantages of sub-volume $v_i$. In this way, we encourage the Glance model $G$ to select sub-volumes with higher advantages within the group, which not only improves the inference efficiency but also reduces false positives by eliminating low-advantage predictions. Ultimately, we optimize the Glance model by maximizing the following objective $\mathcal{J}_{GRL}$:

$$
\begin{aligned}
\mathcal{J}_{GRL}(\theta) &= \mathbb{E}[\{o_i\}_{i=1}^N \sim G(v)] \\
&= \frac{1}{N} \sum_{i=1}^N \left\{ \min[\hat{s}_1 \cdot A_i, \ \hat{s}_2 \cdot A_i] - \beta \mathbb{D}_{KL}[G || G_{ref}] - \alpha \mathbb{CE}(o_i, y_i) \right\}, \\
\mathbb{D}_{KL}(G || G_{ref}) &= \frac{G_{ref}(o_i|v_i)}{G(o_i|v_i)} - \log \frac{G_{ref}(o_i|v_i)}{G(o_i|v_i)} - 1, \\
\hat{s}_1 &= \frac{G(v_i)}{G_{ref}(v_i)}, \quad \hat{s}_2 = \text{clip}\left( \frac{G(v_i)}{G_{ref}(v_i)}, 1 - \epsilon, 1 + \epsilon \right),
\end{aligned}
\tag{4}
$$

where $\theta$ represents the parameters of the Glance model $G$ for updating. $N$ is the number of sub-volumes, $\hat{s}_1$, $\hat{s}_2$, and $\epsilon$ are used to clamp the ratios following previous RL algorithms (Schulman et al., 2017; Shao et al., 2024). $G_{ref}$ represents the reference model (*i.e.*, the initial Glance model). To maintain training stability (Schulman et al., 2017; Shao et al., 2024; Feng et al., 2025), we freeze its parameters and update it at fixed steps following previous RL methods. We also adopt the KL

Table 1: Pan-cancer segmentation performance (DSC %) on internal datasets. Interactive segmentation methods rely on manual prompts, which gain higher performance but cannot be applied to automatic screening. '-' denotes that this model cannot apply to this lesion type.

| Method | Lung tumor | Lung nodule | Pleural effusion | COVID-19 infection | Liver tumor | Pancreas tumor | Kidney tumor | Adreno. carcinoma | Colon tumor | Average (per type) |
|---|---|---|---|---|---|---|---|---|---|---|
| *Interactive segmentation* | | | | | | | | | | |
| *(require manual prompts)* | | | | | | | | | | |
| SegVol (point+text) | 68.9 | - | - | - | 71.8 | - | 68.3 | 90.5 | 69.7 | - |
| SAT (text) | 51.0 | 28.0 | - | 67.2 | 60.1 | 34.2 | 67.9 | - | 35.3 | - |
| ULS (point) | 76.5 | - | - | - | 65.1 | - | 57.9 | 82.9 | 68.3 | - |
| LesionLocator (point) | 77.1 | - | - | - | 76.8 | - | 67.9 | 89.1 | 75.3 | - |
| *Pan-cancer segmentation* | | | | | | | | | | |
| *(automatic screening)* | | | | | | | | | | |
| nnUNet | 50.4 | 48.1 | 36.8 | 61.5 | 61.2 | 35.9 | 68.3 | 86.6 | 30.6 | 53.3 |
| SwinUNETR | 47.8 | 43.6 | 26.3 | 58.5 | 57.0 | 27.6 | 70.8 | 83.0 | 21.0 | 48.6 |
| 3D UX-Net | 42.4 | 30.1 | 10.2 | 56.1 | 54.4 | 16.3 | 62.3 | 82.5 | 6.9 | 40.1 |
| CLIP-driven | 46.7 | 36.8 | 22.8 | 63.1 | 61.2 | 23.7 | 67.0 | 87.5 | 20.2 | 47.6 |
| TransUNet | 51.2 | 36.7 | 17.9 | 60.8 | 59.2 | 29.9 | 64.7 | 87.7 | 14.5 | 46.9 |
| UniMiSS+ | 49.1 | 40.3 | 23.6 | 62.0 | 55.5 | 22.1 | 64.1 | 83.6 | 12.1 | 45.9 |
| VoCo | 53.4 | 49.4 | 41.6 | 64.2 | 65.1 | 36.0 | 70.5 | 89.3 | 34.6 | 56.1 |
| SuPreM | 52.5 | 48.0 | 38.5 | 64.1 | 65.3 | 40.2 | 71.2 | 88.0 | 21.9 | 54.4 |
| PASTA | 52.1 | 48.2 | 23.3 | 64.7 | 66.2 | 30.8 | 72.1 | 88.6 | 29.3 | 52.8 |
| **GF-Screen** | **57.7** | **55.2** | **45.0** | **69.5** | **67.7** | **47.9** | **75.3** | **91.2** | **37.7** | **60.8** |

Table 2: Pan-cancer detection performance (F1-Score %) on internal datasets. We did not compare the interactive-based methods since the prompt will already provide whether this scan contains lesions or not. We report the average results for each CT scan.

| Method | Lung tumor | Lung nodule | Pleural effusion | COVID-19 infection | Liver tumor | Pancreas tumor | Kidney tumor | Adreno. carcinoma | Colon tumor | Average (per type) |
|---|---|---|---|---|---|---|---|---|---|---|
| nnUNet | 92.0 | 86.3 | 92.8 | 96.9 | 95.5 | 85.8 | 97.5 | 93.3 | 72.2 | 90.2 |
| SwinUNETR | 92.5 | 90.3 | 88.9 | 96.2 | 97.8 | 91.9 | 99.5 | 93.3 | 72.0 | 92.5 |
| 3D UX-Net | 89.7 | 87.5 | 84.6 | 97.4 | 95.5 | 88.9 | 97.5 | 65.8 | 35.7 | 86.3 |
| CLIP-driven | 90.2 | 88.2 | 88.9 | 97.4 | 96.7 | 85.8 | 98.8 | 75.5 | 46.7 | 88.1 |
| TransUNet | 90.2 | 87.5 | 79.9 | 96.2 | 95.5 | 87.8 | 99.9 | 93.3 | 64.7 | 89.1 |
| UniMiSS+ | 91.1 | 86.0 | 79.9 | 98.9 | 95.5 | 85.8 | 97.5 | 93.3 | 51.6 | 86.7 |
| VoCo | 91.1 | 86.3 | 92.8 | 96.2 | 97.8 | 91.9 | 98.8 | 93.3 | 82.0 | 92.2 |
| SuPreM | 92.0 | 86.3 | 92.8 | 97.4 | 96.7 | 85.8 | 97.5 | 93.3 | 72.2 | 90.4 |
| PASTA | 89.7 | 86.3 | 79.9 | 96.2 | 96.7 | 85.8 | 97.5 | 93.3 | 72.2 | 88.6 |
| **GF-Screen** | **96.4** | **91.9** | **96.6** | **100.0** | **98.2** | **95.1** | **100.0** | **100.0** | **85.0** | **95.9** |

penalty $\mathbb{D}_{KL}$ to regularize the divergence between $G$ and $G_{ref}$, and $\beta$ denotes the coefficient of $\mathbb{D}_{KL}$. The $\mathbb{D}_{KL}$ term follows prior work (Shao et al., 2024), which employs a second-order Taylor expansion approximation of the standard KL divergence. A detailed derivation is provided in the appendix. We further add the classification loss $\mathbb{CE}$ as Eq. 1 in the objective. Specifically, we employ a small coefficient $\alpha$ to avoid the model overfitting to the negative class as discussed above. Overall, the optimization objective of the Glance model is to maximize the reward, *i.e.*, minimize the loss $L_{GRL} = -\mathcal{J}_{GRL}(\theta)$. Our RL training operates synergistically with segmentation training, enabling an end-to-end framework for pan-cancer screening.

## 4 EXPERIMENTS

### 4.1 EXPERIMENTAL SETTINGS

Our training and validation dataset includes a total of 5,117 scans from 23 public datasets across 9 lesion types, as shown in Appendix Table A1. We set fixed train and val splits for the 16 internal datasets, then aggregate all the training sets in universal training. To demonstrate that the effectiveness of GF-Screen does not rely on dataset-specific biases, we involve 7 external datasets in evaluation, which are unseen in training. For the FLARE25 dataset, we evaluate the performance on the public validation leaderboard, making our work more trustworthy and easier to adopt.

In GF-Screen, we adopt a lightweight 3D ResNet-18 (Chen et al., 2019; He et al., 2016) as the Glance model, and SwinUNETR (Hatamizadeh et al., 2021) as the Focus model for fair comparisons with state-of-the-art models (Liu et al., 2023; Wu et al., 2025a; Li et al., 2024). Thus, SwinUNETR can be

Table 3: Pan-cancer segmentation performance (DSC %) on external datasets. The used external datasets are described in Table A1.

| Method | Rider | Corona | IR. | Avg. |
|---|---|---|---|---|
| nnUNet | 35.3 | 60.7 | 51.0 | 49.0 |
| Swin. | 25.6 | 57.7 | 47.2 | 43.5 |
| UX-Net | 13.5 | 33.6 | 39.4 | 28.8 |
| CLIP. | 27.6 | 50.6 | 51.1 | 43.1 |
| Trans. | 17.6 | 50.6 | 49.0 | 39.1 |
| UniM. | 20.1 | 45.7 | 43.5 | 36.4 |
| VoCo | 31.3 | 55.8 | 56.5 | 47.9 |
| SuPreM | 29.1 | 54.2 | 53.9 | 45.7 |
| PASTA | 24.2 | 58.1 | 52.7 | 45.0 |
| **GF-Screen** | **38.3** | **64.3** | **59.7** | **54.1** |

Table 4: The false positive rates (%) on three internal and external datasets. Lower false positive rates represent higher performance.

| Method | CHAOS | TCIA. | Atlas | Avg. |
|---|---|---|---|---|
| nnUNet | 40.0 | 21.2 | 29.9 | 30.4 |
| Swin. | 40.0 | 41.3 | 61.1 | 47.5 |
| UX-Net | 40.0 | 46.3 | 58.6 | 48.3 |
| CLIP. | 40.0 | 45.0 | 58.6 | 47.9 |
| Trans. | 45.0 | 76.3 | 66.2 | 62.5 |
| UniM. | 55.0 | 65.0 | 63.3 | 61.1 |
| VoCo | 40.0 | 28.8 | 56.5 | 41.8 |
| SuPreM | 40.0 | 32.5 | 43.3 | 38.7 |
| PASTA | 45.0 | 43.8 | 38.9 | 42.6 |
| **GF-Screen** | **10.0** | **13.8** | **22.9** | **15.6** |

seen as the baseline. For the GRL hyperparameters, we empirically set $N = 16$, $\epsilon = 0.1$, $\alpha = 0.1$, and $\beta = 0.01$ following previous GRPO settings. The reference model is updated per epoch, which means that the Glance model trained in the current epoch will serve as the reference model for the next epoch. The coefficients of GRL loss for the Glance model and segmentation loss $L_{seg}$ (typical Dice-CE loss) for the Focus model are set to be equal. The overall loss function is as follows:

$$L = L_{GRL} + L_{seg} = -\mathcal{J}_{GRL}(\theta) + L_{Dice-CE}. \tag{5}$$

All the training experiments are conducted on one NVIDIA A800 (80G) GPU, and the inference can be conducted within 16G GPU storage. We use AdamW as the optimizer with a learning rate of 3e-4, using a cosine decay scheduler. We adopt a batch size of 4, and for each volume, we crop 4 sub-volumes (a total of 16 sub-volumes). In pre-processing, following the settings of the FLARE25 challenge, we split the dataset into chest and abdomen CT scans, then clipped the Hopfield Unit (HU) value (window adjustment) and normalized them to $[0, 1]$ as previous methods. For chest CT, the [min, max] of clipped HU is set to $[-900, 650]$, and $[-175, 250]$ for abdomen CT. The size of each sub-volume is set as $[96, 96, 64]$. During inference, we simply use a sliding-window approach following previous methods. No post-processing techniques are used.

## 4.2 PAN-CANCER SEGMENTATION AND DETECTION

We first evaluate the performance on pan-cancer segmentation as shown in Table 1. Specifically, GF-Screen achieves an average of 60.8% DSC across 9 lesion types, surpassing the second-best method (Wu et al., 2025a) by 4.7%. We also present the results of interactive-based methods. Although these methods can achieve higher performance with the guidance of manual prompts, our model can also obtain competitive results on lung nodules, COVID-19, pancreas tumors, kidney tumors, and adrenocortical carcinoma. The detection F1-Scores are shown in Table 2, where GF-Screen achieves 95.9% F1-Score and surpasses previous methods by a clear margin.

The segmentation results on three external datasets, Rider (Aerts et al., 2014), Corona (Paiva, 2020), and IRCADb (Soler et al., 2010), are shown in Table 3, across lung tumors, COVID-19, and liver tumors. It can be seen that GF-Screen achieves superior performance in external validation, with 54.1% DSC on average and surpasses the second-best nnUNet (Isensee et al., 2021) by 5.1%. We further present the false positive results on three healthy datasets in Table 4. Although some segmentation models like VoCo (Wu et al., 2025a), SuPreM (Li et al., 2024), and PASTA (Lei et al., 2025) can yield competitive segmentation results, they **suffer from high false positives** on healthy datasets. We observe that it is caused by the overhead focusing on redundant healthy regions. GF-Screen is developed to tackle this challenge by discarding low-advantage predictions. For example, compared with SuPreM (Li et al., 2024), GF-Screen achieves 23.1% lower false positives.

## 4.3 SUPERIOR PERFORMANCE ON PAN-CANCER CHALLENGE

We first evaluate on the FLARE23 pan-cancer dataset (Ma et al., 2024) in Table 5. GF-Screen yields 56.7% DSC and surpasses previous methods by a clear margin. We further evaluate the average

Table 5: Performance on FLARE23 dataset.

| Method | DSC |
|---|---|
| nnUNet | 43.7 |
| Swin. | 41.5 |
| UX-Net | 20.1 |
| CLIP. | 28.3 |
| Trans. | 27.2 |
| UniM. | 25.1 |
| VoCo | 50.6 |
| SuPreM | 47.3 |
| PASTA | 34.6 |
| **GF-Screen** | **56.7** |

Table 6: Average inference duration (seconds/per scan).

| Method | Duration |
|---|---|
| nnUNet | 136 |
| Swin. | 114 |
| UX-Net | 102 |
| CLIP. | 109 |
| Trans. | 135 |
| UniM. | 128 |
| VoCo | 114 |
| SuPreM | 114 |
| PASTA | 197 |
| **GF-Screen** | **28** |

Table 7: Evaluation on MICCAI FLARE25 validation leaderboard, compared with the champion solution of FLARE24 (Isensee et al., 2021; Huang, 2024) and several comparison methods.

| Method | DSC | NSD |
|---|---|---|
| *FLARE24 champion* | | |
| nnUNet (Huang, 2024) | 33.0 | 24.0 |
| Swin. | 30.7 | 23.9 |
| VoCo | 47.5 | 41.0 |
| **GF-Screen** | **58.6** | **52.2** |

Table 8: We report the classification sensitivity and specificity of the Glance model.

| Method | Sen. | Spe. | Ratio(%) | GFLOP | DSC |
|---|---|---|---|---|---|
| SwinUNETR | - | - | 100 | 12155 | 48.6 |
| GF-Screen | 97.7 | 75.9 | 16.7 | 2164 | 60.8 |

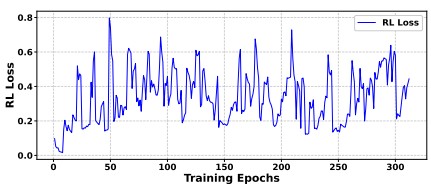

Figure 6: RL loss curve in GRL training.

inference duration results in Table 6, since efficiency plays an important role in the real-world deployment of large-scale cancer screening (Ma et al., 2024). It can be seen that most comparison methods suffer from long inference duration and fail to balance efficiency with accuracy. This is because in these models, huge computation costs will be wasted in redundant regions, while GF-Screen effectively addresses this challenge by discarding redundant regions.

The results on the FLARE25 validation leaderboard are in Table 7. Due to the limitation of submission times, we only compare several comparison methods (Isensee et al., 2021; Wu et al., 2025a; Hatamizadeh et al., 2021). Specifically, Huang (2024) modified nnUNet (Isensee et al., 2021) to balance the efficiency and accuracy, which has won the champion of FLARE24 (Ma et al., 2024). GF-Screen surpasses the champion solution of FLARE24 by a large margin (+25.6% DSC and +28.2% NSD), highlighting a novel and practical breakthrough in pan-cancer screening.

## 4.4 ABLATION STUDIES

We first evaluate GF-Screen in selecting diseased regions and discarding healthy regions. We crop the CT volumes into sub-volumes using a sliding window approach. Then, we evaluate the sensitivity and specificity results of the Glance model in classifying these sub-volumes. On average across

Table 9: Ablation studies of RL training. We report DSC(%) on the FLARE23 dataset. We report the ratio(%) to indicate the proportion of sub-volumes selected by the Glance model.

| Method | DSC | ratio |
|---|---|---|
| SwinUNETR (Hatamizadeh et al., 2021) | 41.5 | 100 |
| *Without RL in Glance model classification training* | | |
| Cross-entropy loss (Eq. 1) | 37.6 | 3.1 |
| Balanced cross-entropy loss | 37.8 | 5.3 |
| Focal loss (Lin et al., 2017) | 36.5 | 4.0 |
| Hard-Negative Sampling (Felzenszwalb et al., 2009) | 35.0 | 3.3 |
| OHEM (Shrivastava et al., 2016) | 39.5 | 7.2 |
| *With RL in Glance model classification training* | | |
| Actor-Critic + PPO (Wang et al., 2020; Huang et al., 2022; Schulman et al., 2017) | 24.5 | 51.6 |
| Group Relative Learning (DSC as rewards) | 43.2 | 21.3 |
| Group Relative Learning (Reward function Eq. 2, without classification loss $\alpha\mathbb{CE}$) | 53.1 | 23.0 |
| Group Relative Learning (Reward function Eq. 2, with classification loss $\alpha\mathbb{CE}$) | 56.7 | 16.7 |

Table 10: Ablation studies of efficiency.

| Method | GFLOP |
|---|---|
| baseline | 12155 |
| GF-Screen | |
| w.o. discard | 13726 |
| w. discard | 2164 |

Table 11: Ablation studies of segmentation backbone.

| Method | DSC |
|---|---|
| UNETR | 16.2 |
| +GF-Screen | 27.3 |
| TransUNet | 27.2 |
| +GF-Screen | 34.8 |
| nnUNet | 43.7 |
| +GF-Screen | 45.6 |
| SwinUNETR | 41.5 |
| +GF-Screen | 56.7 |

Table 12: Ablation studies of the Glance model backbone and $N$.

| Glance model backbone | DSC |
|---|---|
| 3D ResNet-18 (32M) | 56.7 |
| 3D ResNet-34 (63M) | 56.5 |
| 3D UNet (31M) | 56.0 |
| $N$ in group relative learning | DSC |
| $N = 4$ | 45.9 |
| $N = 8$ | 51.6 |
| $N = 16$ | 56.5 |
| $N = 32$ | 56.4 |

all validation datasets, there are 87.6% without lesions and 12.4% with lesions in the sub-volumes. We show the ratio of preserved sub-volumes selected by the Glance model:

$$ratio = \frac{number\ of\ selected\ sub\text{-}volumes}{number\ of\ total\ cropped\ sub\text{-}volumes}. \tag{6}$$

As in Table 8, GF-Screen selects diseased sub-volumes with high sensitivity (97.7%). By discarding 83.3% redundant sub-volumes (preserved ratio 16.7%), GF-Screen reduces the computation cost by an average of 5.7 times. We further analyze the loss curves of GRL, as shown in Fig. 6. We observe that the resulting curves closely align with those in prior GRPO literature, demonstrating the robustness of our training process. More sensitivity analyses of training are in Appendix Fig. A6.

We further conduct ablation studies of RL training in Table 9. We report the results on the FLARE23 pan-cancer dataset. First, as discussed in Section 3.2, we observe that direct usage of cross-entropy loss (Eq. 1) for classification training yields worse results. We observe that the severe foreground-background class imbalance causes the model to collapse into predicting the negative class. As can be seen, the preserved ratio is decreased to 3.1% in this case, which means the Glance model tends to predict the negative class in classification. We observe that simply changing the criterion loss or using hard-sample mining (Felzenszwalb et al., 2009; Shrivastava et al., 2016) cannot effectively improve the performance. We conclude that stronger classification baselines still rely on the coarse volume-level classification labels, which cannot effectively address the foreground-background imbalance. These results motivate us to introduce more precise supervision for training. Thus, we propose to leverage the fine-grained segmentation result as rewards for RL training, which can provide more precise supervision for the Glance model.

Second, we explored the classical PPO algorithm (Schulman et al., 2017) to optimize the Glance model. This approach requires training an additional value model to calculate the rewards. We follow the settings of previous works (Wang et al., 2020; Huang et al., 2022) to train this value model, and optimize the Glance model using the PPO algorithm. However, we observe that the result is bad, and the Glance model cannot stably converge to predict the classes of the sub-volumes. To this end, we highlight that more advanced RL methods should be used.

Third, we evaluate the effectiveness of GRL. We first explore leveraging the DSC results as rewards instead of the binary detection reward in Eq. 2. As shown in Table 9, this approach achieves 43.2% DSC on FLARE23. The performance is slightly better than the baseline, and the efficiency is significantly improved (ratio 21.3%, 78.7% sub-volumes are discarded). We then explore the usage of binary detection reward as Eq. 2, and the performance is further improved. We conclude that for the Glance model, accurate detection is more important. We further add the classification loss $\alpha\mathbb{CE}$ in $\mathcal{J}_{GRL}(\theta)$, which can further improve the performance.

We conduct ablation studies to evaluate the efficiency in Table 10. Although the classification process introduces minor computational overhead, the discarding of redundant healthy regions drastically reduces segmentation costs. We evaluate multiple network architectures in Tables 11 and 12. To balance efficiency and accuracy, we employ 3D ResNet-18 as the Glance model. We further evaluate the settings of $N$ in group relative learning in Table 12. Empirically, we set $N = 16$ in the experiments. **More visualization results are shown in the Appendix.**

## 5 ACKNOWLEDGMENTS

This work was supported by the Hong Kong Innovation and Technology Commission (Project No. GHP/006/22GD and ITCPD/17-9), HKUST (Project No. FS111), HKUST-HKUST(GZ) Cross-Campus Collaborative Research Scheme (Project No. C036), and Guangdong Provincial Department of Science and Technology's '1+1+1' Joint Funding Program for Guangdong-Hong Kong Universities. We highly appreciate the reviewers for their valuable comments on our manuscript, which have significantly improved our manuscript.

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

## 6 APPENDIX

### 6.1 EXPERIMENTAL SETTINGS

Table A1: The datasets used for training and validation in pan-cancer screening. For the Atlas dataset (Bassi et al., 2025), we select the healthy cases based on the provided reports and ensure they do not overlap with other datasets. Overall, there are a total of 5,117 CT scans from 16 internal and 7 external datasets covering 9 different types of lesions.

| Dataset | Tumor/Lesion Types | Scans (Train/Val) |
|---|---|---|
| *Internal datasets* | | |
| MSD06-Lung (Antonelli et al., 2022) | Lung tumor | 46/17 |
| NSCLC-Radiogenomics (Bakr et al., 2018) | Lung tumor | 68/20 |
| NSCLC-Radiomics (Zhang et al., 2017) | Lung tumor | 323/92 |
| LIDC (Armato III et al., 2011) | Lung nodule | 826/184 |
| NSCLC-PleuralEffusion (Arrieta et al., 2019) | Pleural effusion | 63/15 |
| COVID-19 (Roth et al., 2022) | COVID-19 infection | 158/41 |
| MSD03-Liver (Antonelli et al., 2022) | Liver tumor | 95/23 |
| MSD08-Hepatic (Antonelli et al., 2022) | Liver tumor | 248/55 |
| WAWTACE (Bartnik et al., 2024) | Liver tumor | 172/47 |
| HCC (Morshid et al., 2019) | Liver tumor | 60/14 |
| MSD07-Pancreas (Antonelli et al., 2022) | Pancreas tumor | 226/55 |
| PANORAMA (Alves et al., 2024) | Pancreas tumor | 376/106 |
| KiTS23 (Heller et al., 2023) | Kidney tumor | 389/99 |
| Adrenal (Ahmed et al., 2020) | Adrenocortical carcinoma | 44/8 |
| MSD10-Colon (Antonelli et al., 2022) | Colon tumor | 103/23 |
| Atlas (Bassi et al., 2025) | Without lesions | 628/157 |
| *External datasets* | | |
| Rider (Aerts et al., 2014) | Lung tumor | 0/56 |
| Corona (Paiva, 2020) | COVID-19 infection | 0/10 |
| IRCADb (Soler et al., 2010) | Liver tumor | 0/20 |
| CHAOS (Kavur et al., 2021) | Without lesions | 0/20 |
| TCIA-Pancreas (Roth et al., 2016) | Without lesions | 0/80 |
| FLARE23 (Ma et al., 2024) | Pan-cancer | 0/50 |
| FLARE25 (val&test) (Ma et al., 2024) | Pan-cancer | 0/100 |
| **Total** | | **3,825/1,292 (5,117)** |

**Datasets**. We list the used datasets in Table A1. Metadata of the used datasets are shown in Table A2, including scanner, contrast phase, and voxel size. The information about scanners and contrast phases is collected from the public links of the used datasets. Specifically, some public datasets did not disclose scanner manufacturer information or lacked the details of contrast phase information. The leaderboard comparison is conducted under equal conditions. The official FLARE challenge holds a list of datasets for participants to conduct training. No external data is allowed.

**Comparison methods**. We compare state-of-the-art medical image segmentation methods, including nnUNet (Isensee et al., 2021), SwinUNETR (Hatamizadeh et al., 2021), 3D UX-Net (Lee et al., 2023), CLIP-driven (Liu et al., 2023), TransUNet (Chen et al., 2024a), UniMiss+ (Xie et al., 2024), VoCo (Wu et al., 2025a), SuPreM (Li et al., 2024), and PASTA (Lei et al., 2025), which can be leveraged for automatic cancer screening. Specifically, UniMiss+, VoCo, SuPreM, and PASTA are pre-trained models. We fully train all these models on our datasets and report the results. One related work CancerUnit (Chen et al., 2023) is not publicly available, thus we cannot compare it. We further compare several interactive segmentation methods, including SegVol (Du et al., 2024), SAT (Zhao et al., 2025), ULS (de Grauw et al., 2025), and LesionLocator (Rokuss et al., 2025), which require manual prompts like points, boxes, or text descriptions. Note that interactive methods rely on manual prompts, which gain higher performance but cannot be applied to automatic screening. For the interactive methods, we adopt their trained models and evaluate them on our datasets without further

Table A2: Meta information of the used datasets. Some public datasets did not disclose device information or lacked the details of contrast phase information.

| Dataset | Scanner | Contrast |
|---|---|---|
| MSD06-Lung (Antonelli et al., 2022) | - | Contrast-enhanced |
| NSCLC-Radiogenomics (Bakr et al., 2018) | Siemens | Contrast-enhanced |
| NSCLC-Radiomics (Zhang et al., 2017) | Siemens | Contrast-enhanced |
| LIDC (Armato III et al., 2011) | GE,Philips,Toshiba,Siemens | Contrast-enhanced |
| NSCLC-PleuralEffusion (Arrieta et al., 2019) | Siemens | Contrast-enhanced |
| COVID-19 (Roth et al., 2022) | Mixed | Non-contrast |
| MSD03-Liver (Antonelli et al., 2022) | - | Contrast-enhanced |
| MSD08-Hepatic (Antonelli et al., 2022) | - | Contrast-enhanced |
| WAWTACE (Bartnik et al., 2024) | GE,Siemens,Philips,Toshiba | Non-contrast,Arterial,Portal Venous,Delayed |
| HCC (Morshid et al., 2019) | - | Contrast-enhanced |
| MSD07-Pancreas (Antonelli et al., 2022) | - | Contrast-enhanced |
| PANORAMA (Alves et al., 2024) | Toshiba,Siemens | Portal venous |
| KiTS23 (Heller et al., 2023) | Toshiba,Siemens,GE,Philips | Contrast-enhanced |
| Adrenal (Ahmed et al., 2020) | GE,Waukesha,WI,USA | Contrast-enhanced |
| MSD10-Colon (Antonelli et al., 2022) | - | Contrast-enhanced |
| Atlas (Bassi et al., 2025) | Mixed | Multi-contrast phase |
| Rider (Aerts et al., 2014) | Siemens | Contrast-enhanced |
| Corona (Paiva, 2020) | - | Non-contrast |
| IRCADb (Soler et al., 2010) | - | Contrast-enhanced |
| CHAOS (Kavur et al., 2021) | - | Contrast-enhanced |
| TCIA-Pancreas (Roth et al., 2016) | Philips,Siemens | Contrast-enhanced |
| FLARE (Ma et al., 2024) | Siemens,General Electric, Philips, Toshiba,Barco,Vital,PHMS | Plain, artery, portal,delay |

Table A3: Comparison methods in this work.

| Method | Publications |
|---|---|
| *Interactive segmentation (require manual prompts)* | |
| SegVol (Du et al., 2024) | NeurIPS'24 |
| SAT (Zhao et al., 2025) | Npj Dig. Med'25 |
| ULS (de Grauw et al., 2025) | MedIA'25 |
| LesionLocator (Rokuss et al., 2025) | CVPR'25 |
| *Pan-cancer segmentation (automatic screening)* | |
| nnUNet (Isensee et al., 2021) | Nature Methods'21 |
| SwinUNETR (Hatamizadeh et al., 2021) | MICCAIW'21 |
| 3D UX-Net (Lee et al., 2023) | ICLR'23 |
| CLIP-driven (Liu et al., 2023) | ICCV'23 |
| TransUNet (Chen et al., 2024a) | MedIA'24 |
| UniMiSS+ (Xie et al., 2024) | TPAMI'24 |
| VoCo (Wu et al., 2025a) | CVPR'24 |
| SuPreM (Li et al., 2024) | ICLR'24 |
| PASTA (Lei et al., 2025) | arXiv'25 |

finetuning. Since our external datasets are already included in their training, we only compare them on the internal datasets. The comparison methods are listed in Table A3.

**Evaluation.** For segmentation, we report the Dice Similarity Coefficient (DSC) results. For detection, we simply adopt the mask-based detection approach following previous methods (Cao et al., 2023; Hu et al., 2025; Chen et al., 2024b), *i.e.*, once the segmentation prediction overlaps with the ground truth, this case is assumed as detected. We report the F1-Score for each CT scan following (Cao et al., 2023; Hu et al., 2025). We further evaluate the false positive rates on three datasets without lesions, *i.e.*, CHAOS (Kavur et al., 2021), TCIA-Pancreas (Roth et al., 2016), and Atlas (Qu et al., 2023).

Table A4: Pre-processing details and Training settings.

| | |
|---|---|
| Clipped HU for chest CT | [-900, 650] |
| Clipped HU for abdomen CT | [-175, 250] |
| Crop Size | [96, 96, 64] |
| Spacing | [1.0, 1.0, 3.0] |
| Backbone of the Glance model | 3D ResNet18 (Chen et al., 2019) |
| Backbone of the Focus model | SwinUNETR (Hatamizadeh et al., 2021) |
| Network Parameters | 104M (Glance model 32M, Focus model 72M) |
| Glance model Loss | $-\mathcal{J}_{GRL}(\theta)$ (Eq 4) |
| Focus model Loss | Dice-CE |
| Optimizer & Scheduler | AdamW & Cosine |
| Batch size | 16 |
| Learning rate | 3e-4 |
| Training epochs | 300 |

## 6.2 SUPPLEMENTARY RESULTS

Table A5: Pan-cancer segmentation performance (DSC %) on 16 datasets across 9 types of lesions. The internal datasets are described in Table A1. We select the no-lesion cases from Atlas (Bassi et al., 2025). We report the results of automatic screening models.

| Method | MSD06 | Radiogenomics | Radiomics | LIDC | Pleural. | COVID-19 | MSD03 | MSD08 |
|---|---|---|---|---|---|---|---|---|
| nnUNet | 48.7 | 61.0 | 48.4 | 48.1 | 36.8 | 61.5 | 57.7 | 53.6 |
| SwinUNETR | 48.7 | 71.0 | 42.6 | 43.6 | 26.3 | 58.5 | 48.5 | 47.9 |
| 3D UX-Net | 45.3 | 70.8 | 35.6 | 30.1 | 10.2 | 56.1 | 51.4 | 45.2 |
| CLIP-driven | 50.0 | 71.7 | 40.6 | 36.8 | 22.8 | 63.1 | 59.7 | 57.0 |
| TransUNet | 54.9 | 69.4 | 46.5 | 36.7 | 17.9 | 60.8 | 48.8 | 57.2 |
| UniMiSS+ | 53.1 | 65.8 | 44.7 | 40.3 | 23.6 | 62.0 | 57.0 | 45.5 |
| VoCo | 54.2 | 71.8 | 49.3 | 49.4 | 41.6 | 64.2 | 63.4 | 59.5 |
| SuPreM | 57.9 | 68.2 | 48.0 | 48.0 | 38.5 | 64.1 | 62.5 | 55.1 |
| PASTA | 56.0 | 64.3 | 48.7 | 48.2 | 23.3 | 64.7 | 58.5 | 59.2 |
| **GF-Screen** | **62.8** | **73.6** | **53.3** | **55.2** | **45.0** | **69.5** | **63.4** | **59.5** |
| | WAWTACE | HCC | MSD07 | PANORAMA | KiTS23 | Adrenal | MSD10 | Atlas |
| nnUNet | 68.4 | 72.8 | 42.3 | 32.6 | 68.3 | 86.6 | 30.6 | 92.6 |
| SwinUNETR | 67.8 | 70.5 | 30.1 | 26.3 | 70.8 | 83.0 | 21.0 | 84.1 |
| 3D UX-Net | 63.3 | 65.5 | 25.4 | 11.6 | 62.3 | 82.5 | 6.9 | 83.3 |
| CLIP-driven | 63.9 | 70.7 | 40.8 | 14.8 | 67.0 | 87.5 | 20.2 | 87.1 |
| TransUNet | 63.7 | 68.8 | 34.7 | 27.4 | 64.7 | 87.7 | 14.5 | 85.8 |
| UniMiSS+ | 61.4 | 68.5 | 29.9 | 18.0 | 64.1 | 83.6 | 12.1 | 90.4 |
| VoCo | 75.6 | 80.2 | 41.4 | 33.2 | 70.5 | 89.3 | 34.6 | 92.6 |
| SuPreM | 75.3 | 76.2 | 45.4 | 37.5 | 71.2 | 88.0 | 21.9 | 92.4 |
| PASTA | 75.1 | 76.1 | 29.2 | 31.6 | 72.1 | 88.6 | 29.3 | 91.5 |
| **GF-Screen** | **75.6** | **80.2** | **51.8** | **45.9** | **75.3** | **91.2** | **37.7** | **93.2** |

Table A6: We conduct five experiments for GF-Screen, and report the mean and standard deviation (std) of segmentation DSC across 9 lesion types.

| Method | Lung tumor | Lung nodule | Pleural effusion | COVID-19 infection | Liver tumor | Pancreas tumor | Kidney tumor | Adreno. carcinoma | Colon tumor |
|---|---|---|---|---|---|---|---|---|---|
| GF-Screen | $57.5_{\pm0.8}$ | $55.4_{\pm0.3}$ | $45.0_{\pm0.3}$ | $69.0_{\pm0.7}$ | $67.6_{\pm0.5}$ | $49.2_{\pm0.6}$ | $75.3_{\pm0.3}$ | $91.5_{\pm1.1}$ | $38.1_{\pm1.2}$ |

Table A7: We conduct five experiments for GF-Screen, and report the mean and standard deviation (std) of detection F1-Score across 9 lesion types.

| Method | Lung tumor | Lung nodule | Pleural effusion | COVID-19 infection | Liver tumor | Pancreas tumor | Kidney tumor | Adreno. carcinoma | Colon tumor |
|---|---|---|---|---|---|---|---|---|---|
| GF-Screen | $96.4_{\pm0.2}$ | $91.9_{\pm0.1}$ | $96.6_{\pm0.1}$ | $100.0_{\pm0.0}$ | $98.2_{\pm0.2}$ | $95.1_{\pm0.2}$ | $100.0_{\pm0.0}$ | $100.0_{\pm0.0}$ | $85.0_{\pm0.5}$ |

Table A8: The t-test p-values compared with the best-competing method VoCo (Wu et al., 2025a).

| P-values | Lung tumor | Lung nodule | Pleural effusion | COVID-19 infection | Liver tumor | Pancreas tumor | Kidney tumor | Adreno. carcinoma | Colon tumor |
|---|---|---|---|---|---|---|---|---|---|
| Segmentation DSC | $4.54 \times 10^{-3}$ | $3.01 \times 10^{-4}$ | $7.46 \times 10^{-5}$ | $3.34 \times 10^{-3}$ | $8.99 \times 10^{-3}$ | $3.35 \times 10^{-4}$ | $7.76 \times 10^{-3}$ | $9.01 \times 10^{-4}$ | $4.89 \times 10^{-3}$ |
| Detection F1-scores | $1.80 \times 10^{-3}$ | $2.06 \times 10^{-5}$ | $4.11 \times 10^{-3}$ | $2.54 \times 10^{-2}$ | $1.53 \times 10^{-2}$ | $9.13 \times 10^{-3}$ | $8.10 \times 10^{-2}$ | $1.15 \times 10^{-2}$ | $6.80 \times 10^{-4}$ |

Table A9: We conduct ablation studies on the sliding window sizes across different types of lesions.

| Sliding window size | Lung tumor | Lung nodule | Pleural effusion | COVID-19 infection | Liver tumor | Pancreas tumor | Kidney tumor | Adreno. carcinoma | Colon tumor |
|---|---|---|---|---|---|---|---|---|---|
| $64 \times 64 \times 64$ | 54.6 | 53.8 | 44.1 | 68.2 | 64.5 | 46.0 | 71.8 | 90.2 | 34.3 |
| $96 \times 96 \times 64$ | 57.5 | 55.4 | 45.0 | 69.0 | 67.6 | 49.2 | 75.3 | 91.5 | 38.1 |
| $128 \times 128 \times 64$ | 57.8 | 55.6 | 42.4 | 69.1 | 67.0 | 49.5 | 74.4 | 91.4 | 38.9 |

We present the dataset-wise results on 16 datasets as a complement to Table 1, as shown in Table A5. To evaluate the training deviation of GF-Screen, we conduct five experiments for GF-Screen, and report the mean and standard deviation (std) in Tables A6 and A7. We further summarize the comparisons with the second-best model (Wu et al., 2025a) in Fig. A1. We also conduct ablation studies on the sizes of the sliding window, as shown in Table A9. We set the size of the sliding window as $96 \times 96 \times 64$ to balance the performance and inference efficiency. We observe that $64 \times 64 \times 64$ is worse, while the results of $96 \times 96 \times 64$ and $128 \times 128 \times 64$ are both competitive.

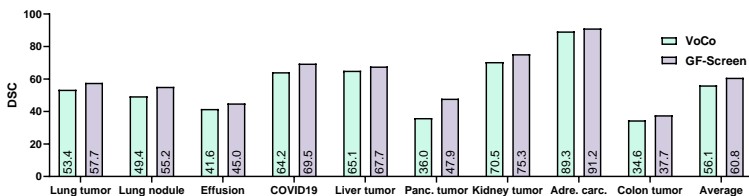

Figure A1: We further summarize the comparisons with the second-best model (Wu et al., 2025a). GF-Screen is 5.7 times faster with higher performance simultaneously.

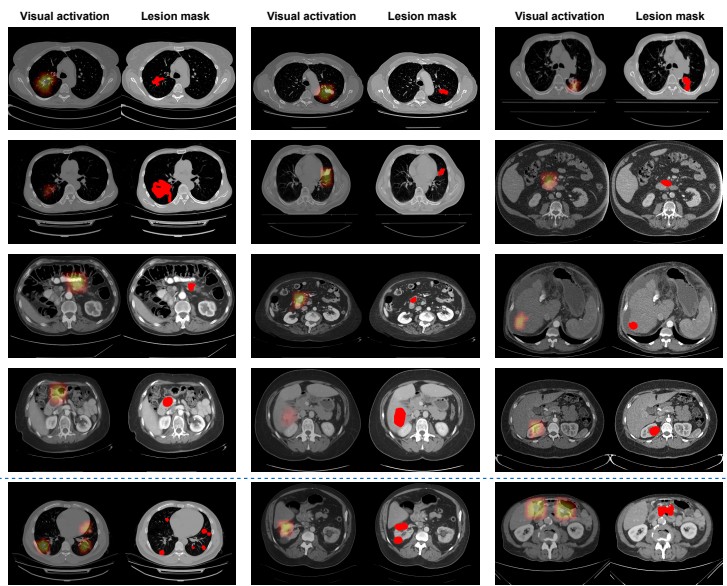

Figure A2: We present the visual activation maps of the Glance model compared with the Ground truth lesion masks. In the last row, we show some failure cases. For example, in the first case of the last row, the model misses the tiny lesion in the left upper part.

## 6.3  VISUALIZATION ANALYSIS OF THE GLANCE MODEL

We analyze the classification results of the Glance model in Fig. A2. We present the visual activation maps of the Glance model compared with the ground truth of lesions. It can be seen that the Glance model can effectively detect the lesions at a coarse sub-volume level, providing optimal diagnostic views for the Focus model for segmentation. We show some failure cases in the last row, where the Glance model misses tiny lesions or fails to indicate the precise regions.

## 6.4 REDUCE FALSE POSITIVES

We observe that previous methods (Wu et al., 2025a; Li et al., 2024; Lei et al., 2025; Hatamizadeh et al., 2021; Liu et al., 2023; Xie et al., 2024; Lee et al., 2023) generally gain high false positives on healthy regions (Table 4), where our GF-Screen can effectively address this challenge by discarding the redundant regions during inference, as shown in Fig. A3. By reducing 23.1% false positives, GF-Screen can also effectively improve the overall DSC scores.

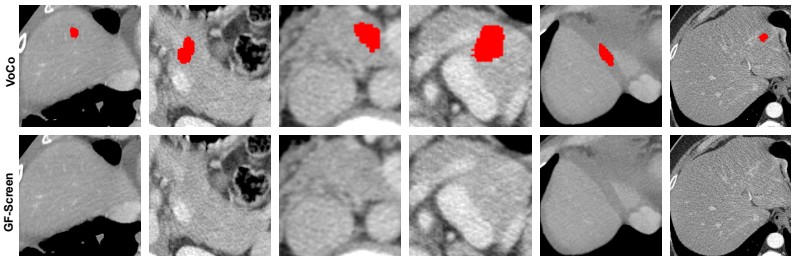

Figure A3: We present the segmentation results on healthy regions (no lesions), compared with the best-competing method (Wu et al., 2025a).

## 6.5 EFFECTIVENESS OF SELECTING OPTIMAL DIAGNOSIS VIEWS

GF-Screen can improve the segmentation DSC by selecting the optimal diagnosis views, as shown in Fig. A4. The optimal view (in red) contains the complete organ and provides important information (*e.g.*, intensity contrast) for cancer diagnosis, while the challenging view (in blue) contains partial information. Thus, the segmentation predictions of this optimal view would be more accurate. Following previous methods, we adopt sliding-window inference to crop sub-volumes with overlapped areas. However, during sliding-window inference, previous methods generally average these results, where the segmentation predictions from challenging views can degrade the average performance of the final results. Our GF-Screen can effectively select optimal views and discard the suboptimal views, mitigating their negative influence.

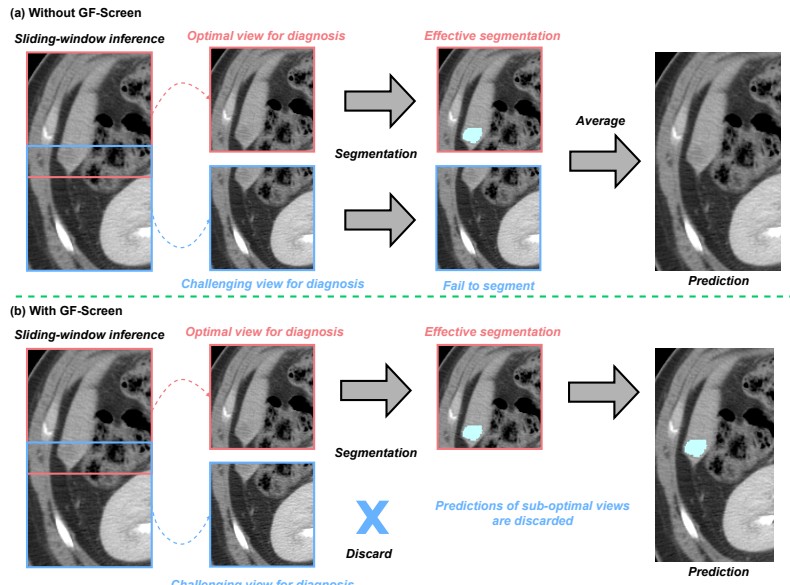

Figure A4: An illustrative case to demonstrate the effectiveness of selecting optimal views. (a) Without GF-Screen, segmentation predictions from challenging views can degrade the average performance of the final results. (b) With GF-Screen, only the segmentation results from optimal diagnostic views are retained, while predictions from suboptimal views are filtered out.

## 6.6 SEGMENTATION RESULTS

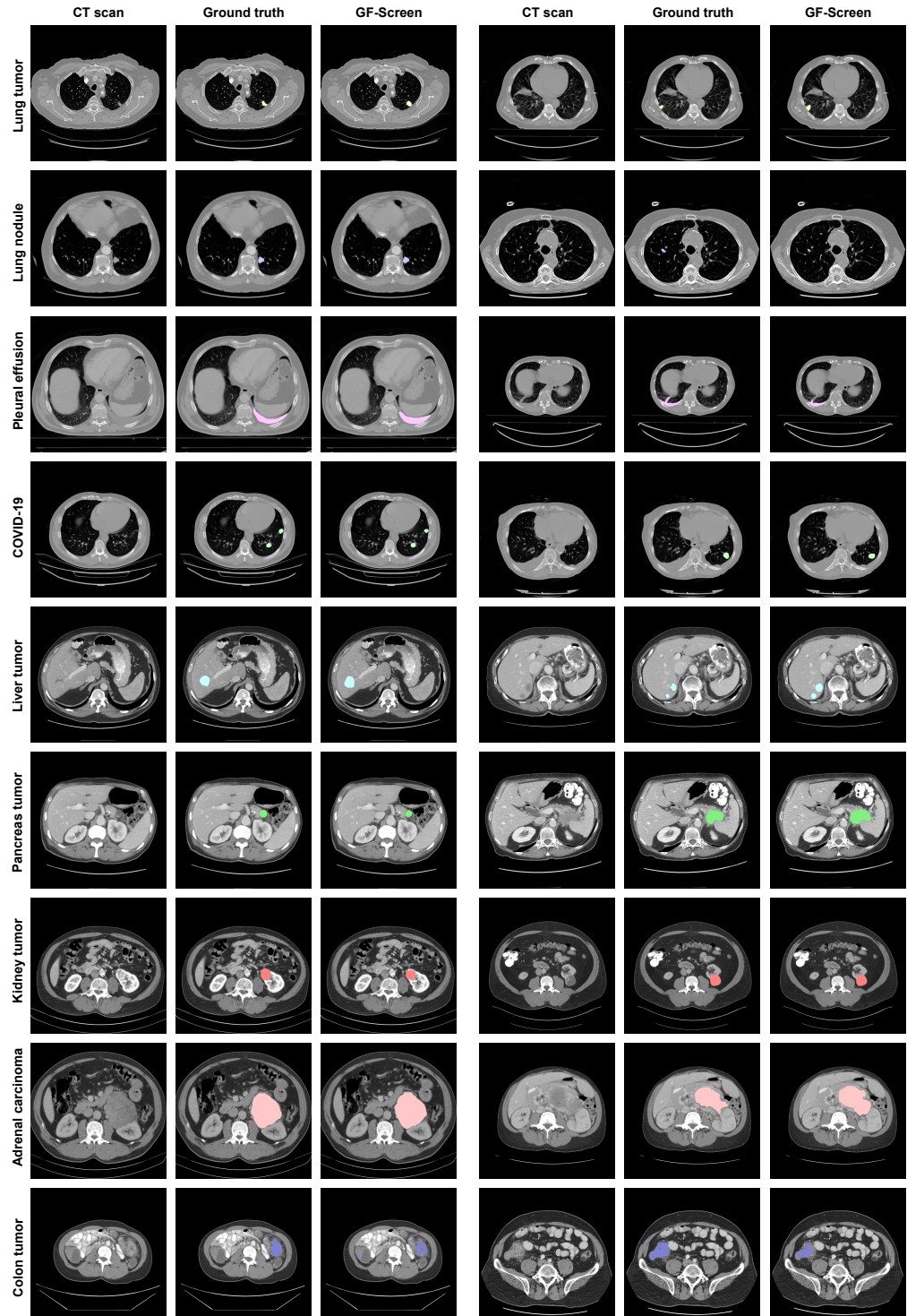

Figure A5: We present the segmentation results of GF-Screen across different types of lesions.

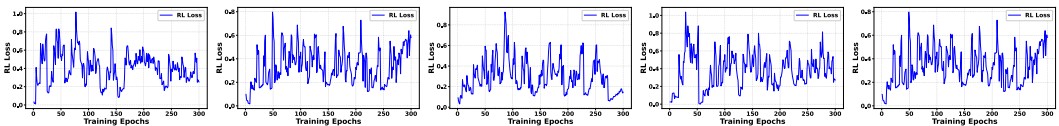

Figure A6: We randomly split our training dataset into five subsets and perform training to analyze the loss curves of RL.

## 6.7 DERIVATION OF THE KL DIVERGENCE APPROXIMATION

In Eq. 4, we utilize a specific form for the Kullback-Leibler (KL) divergence regularization term $\mathbb{D}_{KL}$, which originates from the GRPO algorithm (Shao et al., 2024). This form is an approximation of the standard reverse KL divergence, derived via a second-order Taylor expansion. Here, we provide the detailed derivation for clarity.

The standard reverse KL divergence between two distributions $G$ and $G_{ref}$ for a given input $v_i$ is defined as:

$$\mathbb{D}_{KL}^{\text{standard}}(G||G_{ref}) = \mathbb{E}_{o_i \sim G(\cdot|v_i)} \left[ \log \frac{G(o_i|v_i)}{G_{ref}(o_i|v_i)} \right]. \tag{7}$$

Let us define the ratio $r(o_i) = \frac{G_{ref}(o_i|v_i)}{G(o_i|v_i)}$. We can re-express the log term as:

$$\log \frac{G(o_i|v_i)}{G_{ref}(o_i|v_i)} = -\log r(o_i). \tag{8}$$

The standard KL divergence then becomes:

$$\mathbb{D}_{KL}^{\text{standard}}(G||G_{ref}) = \mathbb{E}_{o_i \sim G} \left[ -\log r(o_i) \right]. \tag{9}$$

Now, consider the function $f(r) = -\log(r)$. We perform a second-order Taylor expansion of $f(r)$ around $r = 1$ (which corresponds to the point where $G$ and $G_{ref}$ are identical).

$$f(r) \approx f(1) + f'(1)(r-1) + \frac{1}{2}f''(1)(r-1)^2 \tag{10}$$

$$= -\log(1) + \left(-\frac{1}{1}\right)(r-1) + \frac{1}{2}\left(\frac{1}{1^2}\right)(r-1)^2 \tag{11}$$

$$= 0 - (r-1) + \frac{1}{2}(r-1)^2 \tag{12}$$

$$= -r + 1 + \frac{1}{2}(r^2 - 2r + 1) \tag{13}$$

$$= \frac{1}{2}r^2 - 2r + \frac{3}{2}. \tag{14}$$

Its second-order Taylor expansion around $r = 1$ is:

$$g(r) \approx g(1) + g'(1)(r-1) + \frac{1}{2}g''(1)(r-1)^2 \tag{15}$$

$$= (1 - \log 1 - 1) + \left(1 - \frac{1}{1}\right)(r-1) + \frac{1}{2}\left(\frac{1}{1^2}\right)(r-1)^2 \tag{16}$$

$$= 0 + 0 \cdot (r-1) + \frac{1}{2}(r-1)^2 \tag{17}$$

$$= \frac{1}{2}(r-1)^2. \tag{18}$$

Notice that $g(r) = r - \log r - 1$ is always non-negative and has a minimum of 0 at $r = 1$, sharing key properties with the KL divergence. More importantly, its expectation $\mathbb{E}_{o_i \sim G}[g(r(o_i))]$ recovers

the form used in our objective:

$$\mathbb{E}_{o_i \sim G}\left[g\left(\frac{G_{ref}(o_i|v_i)}{G(o_i|v_i)}\right)\right] = \mathbb{E}_{o_i \sim G}\left[\frac{G_{ref}(o_i|v_i)}{G(o_i|v_i)} - \log \frac{G_{ref}(o_i|v_i)}{G(o_i|v_i)} - 1\right] \quad (19)$$

$$= \mathbb{D}_{KL}(G||G_{ref}) \quad \text{(as defined in Eq. 4).} \quad (20)$$

This form, $r - \log r - 1$, is preferred in policy optimization algorithms like GRPO because it serves as a principled, computationally friendly surrogate loss. It penalizes large deviations of the policy $G$ from the reference $G_{ref}$ (i.e., when $r$ deviates from 1), helping to ensure stable training, while being easier to optimize in practice than the exact KL divergence.

## 6.8 THE USE OF LARGE LANGUAGE MODELS (LLMS)

We use LLMs (DeepSeek) only to check whether there are grammar errors.

## 6.9 REPRODUCIBILITY STATEMENT

**We promise that our code and model checkpoints will be released**. We have submitted our Docker image with our code and model checkpoints for public leaderboard validation, ensuring our results are trustworthy and easy to adopt.

## 6.10 FUTURE DIRECTIONS

Our main contribution lies in developing the first RL framework for tackling the challenges in pan-cancer screening, while the adaptation of GRPO only constitutes the optimization part of the whole framework. Our framework can potentially integrate more emerging policy optimization algorithms beyond GRPO in the future. We will further involve more datasets and lesion types in training and validation, *e.g.*, including head-neck CT datasets for lymph node detection. And we will explore combining the classification of different lesion types into our framework. In the future, we will further explore the usage of GRL to advance more medical image analysis tasks.

Although GF-Screen has achieved superior performance on 23 public datasets, further exploration of its application to real-world datasets is necessary to substantiate its effectiveness in clinical practice. In the future, we will also include clinician-in-the-loop evaluation and human–AI comparison to strengthen the evaluation of our method. We will work with our collaborating hospitals, *e.g.*, conduct a reader study using real-world data. We will compare the diagnostic efficiency and accuracy of radiologists with and without AI assistance from GF-Screen.

## 6.11 CONCLUSION

We introduce GF-Screen, a Glance and Focus RL framework for tackling the challenges in pan-cancer screening. GF-Screen employs a Glance model to select diseased regions and discard redundant healthy regions, where the diseased regions are fed into the Focus model for precise segmentation. We innovatively leverage the segmentation results of the Focus model to reward the Glance model via RL, effectively encouraging the Glance model to prioritize high-advantage predictions. Extensive experiments on 16 internal and 7 external datasets across 9 lesion types demonstrated the superior performance of GF-Screen. In summary, our work did not aim to propose a new theory for RL. Instead, our contribution is introducing the first RL framework specifically designed for pan-cancer screening, which is a novel and practical conceptual breakthrough in this field.

