# OpenReview forum: "Glance and Focus Reinforcement for Pan-cancer Screening"
_ICLR.cc/2026/Conference — ICLR 2026 Poster_

### Official Review · Reviewer_Cyrm · 2025-10-26

**Soundness:** 4
**Presentation:** 4
**Contribution:** 4
**Rating:** 10
**Confidence:** 5

**Summary:**

This paper presents a novel end-to-end reinforcement learning framework for cancer screening. The proposed method consists of a sliding window-based approach (glance) to first identify CT sub volumes and a focus model to segment the lesions. This glance-and-focus approach demonstrated superior performance on multiple internal and public datasets, as well as the Miccai FLARE25 challenge.

**Strengths:**

1. The idea of applying group relative learning for sub-volume selection is innovative. Moreover, the reward design that incorporates segmentation results as reinforcement feedback is well-motivated. This strategy effectively distinguishes sub-volumes with partial lesion overlap, enabling more accurate and meaningful reward signals for reinforcement learning.

2. The proposed method is fast in inference by discarding redundant regions, which reduces computational cost.

3. The experimental evaluation is comprehensive, including comparisons with multiple strong baselines and demonstrating consistent superior performance across all tasks.

**Weaknesses:**

1. Equation 4 has some jump in thoughts. I understand that it comes from the second order Taylor expansion of KL divergence and serves as a surrogate of KL divergence. However, it is not straightforward to readers. I recommend explaining this jump in the supplementary materials.

2.The evaluation datasets only contain abdominal CT images. It is unclear how the model performs on other parts of the body, e.g., head CT.

3. It would be beneficial to the paper to present the protocols and device of the CT images in use. This could benefit the robustness of the proposed method.

**Questions:**

1. How is the size of sliding window decided? Different tumors, for example, liver tumor and lung tumor, differentiate in size. For the same type of tumor, it could also have inter-patient variations.

2. Did the author perform any statistical analysis of the results in Tables 2 and 3?

---

> ### Author Response · Authors · 2025-11-20
> **[1/2] Response to Reviewer Cyrm**
>
> We are truly honored and greatly encouraged by your heartening recognition of our work! We deeply appreciate you for providing such valuable comments. We have carefully studied your comments and have revised our manuscript accordingly. Our point-by-point responses are detailed below.
> ___
>
> **Comment 1**: Equation 4 has some jump in thoughts...I recommend explaining this jump in the supplementary materials.
>
> **Response 1**: We sincerely appreciate your careful review and valuable suggestion. We have added the detailed derivation of the KL Divergence in Appendix Section 6.7. We have also added a note in the main paper as follows:
>
> *"The D_KL term employs a second-order Taylor expansion approximation of the standard KL divergence. A detailed derivation is provided in the appendix."*
>
> ___
>
> **Comment 2**: The evaluation datasets only contain abdominal CT images. It is unclear how the model performs on other parts of the body, e.g., head CT.
>
> **Response 2**: Thanks for your valuable comment. First, we would like to clarify that our dataset contain both **chest and abdominal CT**, where the chest CT datasets cover lung tumors, lung nodules, COVID-19 infection, and pleural effusion.
>
> For the head region, CT is generally used for trauma or fracture screening, while MRI is the primary and commonly-used imaging tool for brain tumors. In the future, we will further involve head-neck CT datasets for lymph node detection. Following your valuable suggestion, we have highlighted this part in the revision (Section 6.10 Future Directions):
>
> *"We will further involve more datasets and lesion types in training and validation, e.g., including head-neck CT datasets for lymph node detection."*
>
> ___
>
> **Comment 3**: It would be beneficial to the paper to present the protocols and device of the CT images in use. This could benefit the robustness of the proposed method.
>
> **Response 3**: Thank you for providing this valuable suggestion. In the revision (Table A2), we have presented the devices and contrast phases for the used datasets as follows, while some public datasets did not disclose details of the protocols.
>
> | Dataset | Scanner | Contrast |
> |---------|---------|----------|
> | MSD06-Lung | - | Contrast-enhanced |
> | NSCLC-Radiogenomics | Siemens | Contrast-enhanced |
> | NSCLC-Radiomics | Siemens | Contrast-enhanced |
> | LIDC | GE,Philips,Toshiba,Siemens | Contrast-enhanced |
> | NSCLC-PleuralEffusion | Siemens | Contrast-enhanced |
> | COVID-19 | Mixed | Non-contrast |
> | MSD03-Liver | - | Contrast-enhanced |
> | MSD08-Hepatic | - | Contrast-enhanced |
> | WAWTACE | GE,Siemens,Philips,Toshiba | Non-contrast,Arterial,Portal Venous,Delayed |
> | HCC | - | Contrast-enhanced |
> | MSD07-Pancreas | - | Contrast-enhanced |
> | PANORAMA | Toshiba,Siemens | Portal venous |
> | KiTS23 | Toshiba,Siemens,GE,Philips | Contrast-enhanced |
> | Adrenal | GE,Waukesha,WI,USA | Contrast-enhanced |
> | MSD10-Colon | - | Contrast-enhanced |
> | Atlas | Mixed | Multi-contrast phase |
> | Rider | Siemens | Contrast-enhanced |
> | Corona | - | Non-contrast |
> | IRCADb | - | Contrast-enhanced |
> | CHAOS | - | Contrast-enhanced |
> | TCIA-Pancreas | Philips,Siemens | Contrast-enhanced |
> | FLARE | Siemens,General Electric, Philips, Toshiba,Barco,Vital,PHMS | Plain, artery, portal,delay |

---

> ### Author Response · Authors · 2025-11-20
> **[2/2] Response to Reviewer Cyrm**
>
> **Comment 4**: The size of sliding window: Different tumors differentiate in size. It could also have inter-patient variations.
>
> **Response 4**: Thanks for your valuable concern. Since our pan-cancer screening task requires to develop one universal model for various types of lesions, we cannot pre-define specific window sizes for each input CT scan. Currently, we set the size of sliding window as *96x96x64* to **balance the performance and inference efficiency**. Following your valuable suggestion, we have added an ablation study of the sliding window size in the revision (Table A9) as follows. We observe that for different types of lesions, *64x64x64* is worse, while the results of *96x96x64* and *128x128x64* are both competitive.
>
> | Sliding window size  | Lung tumor | Lung nodule | Pleural effusion | COVID-19 infection | Liver tumor | Pancreas tumor | Kidney tumor | Adreno. carcinoma | Colon tumor |
> |-----------|------------|-------------|------------------|-------------------|-------------|---------------|--------------|------------------|-------------|
> | 64×64×64           | 54.6 | 53.8 | 44.1 | 68.2 | 64.5 | 46.0 | 71.8 | 90.2 | 34.3 |
> | 96×96×64           | 57.5 | 55.4 | 45.0 | 69.0 | 67.6 | 49.2 | 75.3 | 91.5 | 38.1 |
> | 128×128×64         | 57.8 | 55.6 | 42.4 | 69.1 | 67.0 | 49.5 | 74.4 | 91.4 | 38.9 |
>
> ___
>
> **Comment 5**: Did the author perform any statistical analysis of the results in Tables 2 and 3?
>
> **Response 5**: Thanks a lot for your valuable concern, ensuring the rigor of our experiments. Previously, we only report the statistical analysis of the segmentation performance. Following your valuable suggestion, we have further added the statistical analysis of detection F1-Score in the revision. The results are as follows (Tables A6 and A7 in the paper), we observe that the training of GF-Screen is stable and achieve competitive results consistently. We have further added the t-test p-values compared with the best-competing method in the revision (Table A8).
>
> | Method    | Lung tumor | Lung nodule | Pleural effusion | COVID-19 infection | Liver tumor | Pancreas tumor | Kidney tumor | Adreno. carcinoma | Colon tumor |
> |-----------|------------|-------------|------------------|-------------------|-------------|---------------|--------------|------------------|-------------|
> | GF-Screen | 57.5±0.8   | 55.4±0.3    | 45.0±0.3         | 69.0±0.7          | 67.6±0.5    | 49.2±0.6      | 75.3±0.3     | 91.5±1.1         | 38.1±1.2    |
>
> | Method    | Lung tumor | Lung nodule | Pleural effusion | COVID-19 infection | Liver tumor | Pancreas tumor | Kidney tumor | Adreno. carcinoma | Colon tumor |
> |-----------|------------|-------------|------------------|--------------------|-------------|----------------|--------------|-------------------|-------------|
> | GF-Screen | 96.4±0.2   | 91.9±0.1    | 96.6±0.1         | 100.0±0.0          | 98.2±0.2    | 95.1±0.2       | 100.0±0.0    | 100.0±0.0         | 85.0±0.5    |
>
> | P-values       | Lung tumor | Lung nodule | Pleural effusion | COVID-19 infection | Liver tumor | Pancreas tumor | Kidney tumor | Adreno. carcinoma | Colon tumor |
> |----------------|------------|-------------|------------------|--------------------|-------------|----------------|--------------|------------------|-------------|
> | Segmentation DSC | 4.54×10⁻³ | 3.01×10⁻⁴ | 7.46×10⁻⁵ | 3.34×10⁻³ | 8.99×10⁻³ | 3.35×10⁻⁴ | 7.76×10⁻³ | 9.01×10⁻⁴ | 4.89×10⁻³ |
> | Detection F1-scores| 1.80×10⁻³ | 2.06×10⁻⁵ | 4.11×10⁻³ | 2.54×10⁻² | 1.53×10⁻² | 9.13×10⁻³ | 8.10×10⁻² | 1.15×10⁻² | 6.80×10⁻⁴ |
>
> Thank you again for your thorough review and valuable suggestions!

---

> ### Comment · Reviewer_Cyrm · 2025-11-27
>
> There exists prior research on mimicking radiologists' radiograph reading behavior for lesion localization using reinforcement learning. I recommend citing this work as a previous contribution to the field.
> Reference:
> Hao, Degan, Dooman Arefan, and Shandong Wu. "Incorporate radiograph-reading behavior and knowledge into deep reinforcement learning for lesion localization." Medical Imaging 2022: Computer-Aided Diagnosis. Vol. 12033. SPIE, 2022.

---

> > ### Author Response · Authors · 2025-11-27
> > **Response from authors**
> >
> > Thanks a lot for suggesting this valuable work! We have cited this work in the revision (Section 2.2) as follows:
> >
> > *"In medical image analysis, Hao et al. proposed an RL method to imitate human visual search behavior for lesion localization in X-ray images."*
> >
> > Reference: Degan Hao, Dooman Arefan, and Shandong Wu. Incorporate radiograph-reading behavior and
> > knowledge into deep reinforcement learning for lesion localization. In Medical Imaging 2022:
> > Computer-Aided Diagnosis, volume 12033, pp. 258–263. SPIE, 2022.
> >
> > Thank you again for your prompt feedback! Please let us know if there are any other points you would like us to address.

---

### Official Review · Reviewer_bZ4d · 2025-10-27

**Soundness:** 2
**Presentation:** 3
**Contribution:** 2
**Rating:** 4
**Confidence:** 4

**Summary:**

This work presents GF-Screen, a Glance-and-Focus reinforcement learning framework for pan-cancer screening in CT scans, inspired by radiologists’ diagnostic workflow. A Glance model identifies suspicious sub-volumes, and a Focus model segments lesions, with segmentation results rewarding the Glance model via a group relative learning strategy. Trained on 5,117 CT scans and evaluated across 23 datasets (9 lesion types), GF-Screen leads the MICCAI FLARE25 leaderboard, surpassing the prior champion by +25.6% DSC and +28.2% NSD, while reducing computation by 5.7× through efficient region selection. This marks a practical step in accurate, scalable, and efficient AI-based pan-cancer screening.

**Strengths:**

1. The authors effectively reframe the well-established coarse-to-fine paradigm as a "glance-and-focus" strategy, an evocative and clinically intuitive framing that demonstrates strong scientific storytelling.
2. The figures are clean, well-designed, and complemented by clear, concise textual descriptions, making the methodology easy to follow.
3. The empirical evaluation is extensive, with experiments across multiple internal and external datasets and diverse lesion types, providing thorough support for the paper’s claims.

**Weaknesses:**

1. The motivation for using reinforcement learning (RL) to train the Glance model is underdeveloped. While extreme class imbalance is acknowledged as a key challenge, the authors do not sufficiently justify why conventional approaches, such as intelligent sampling, focal loss, or hard-negative mining, would be inadequate. Jumping directly to RL, a currently popular but complex paradigm, feels more like a methodological trend than a necessity.
2. Within the RL framework, using segmentation performance from the Focus model as the reward signal for the Glance model introduces undesirable coupling between the two stages. Specifically, a high segmentation score may simply reflect an "easy" sub-volume (e.g., one with clear boundaries or canonical orientation), not necessarily one that was optimally cropped. Conversely, a low segmentation score might indicate a challenging but clinically critical case (e.g., partial lesions or suboptimal viewing angles), precisely the cases the Glance model should prioritize. The current reward scheme risks biasing the Glance model toward easy regions and discarding hard, ambiguous, yet important ones.
3. The paper’s layout could be better optimized for impact. Given space constraints, Table 1 (dataset statistics) could be moved to the supplementary material to free up room for more critical content, particularly ablation studies and a dedicated conclusion section, which is notably absent and atypical for a conference submission.
4. From a methodological standpoint, the work primarily constitutes an application of the GRPO algorithm to a new domain, rather than a significant algorithmic advance.

**Questions:**

1. The term "data curation" is used, but simply aggregating public datasets does not constitute curation. Could the authors clarify their curation process? Specifically, how were label inconsistencies or annotation discrepancies across source datasets identified and resolved?
2. Regarding the FLARE25 challenge results: the reported margin over the second-place method is substantial. Was this comparison conducted under equal conditions? In particular, did all participants use the same training data? Does the challenge permit the use of external data?

---

> ### Author Response · Authors · 2025-11-20
> **[1/3] Response to Reviewer bZ4d**
>
> We are deeply grateful for your professional insights and constructive suggestions to improve our work. We have meticulously studied your valuable comments and thoroughly revised our manuscript accordingly. Our detailed responses are as follows.
>
> ___
>
> **Comment 1**: Motivation for using RL: sufficiently justify why conventional approaches would be inadequate ... Jumping directly to RL ... feels more like a methodological trend than a necessity.
>
> **Response 1**: Thanks for your valuable comment, making our motivation clearer. We strongly resonate with your research taste. We would like to highlight that **the adoption of RL is driven by the specific challenges of our task**, but not chasing the methodological trend. To lead the challenge leaderboard, **practicality and performance are our priority**.
>
> First, we would like to clarify that we did not jump directly to RL but started from the conventional classification approaches as discussed in Section 3.2.
> In response to your valuable consideration, we would like to discuss this important issue from two aspects: (1) why conventional classification approaches are inadequate; (2) why adopt RL in our task. Concretely:
>
> **[Conventional classification approaches]** In Section 3.2 (Line 198-237), we thoroughly analyzed the two fundamental shortcomings of conventional approaches: (1) foreground-background imbalance results in negative overfitting and (2) lesion-containing sub-volumes while with suboptimal views deteriorate classification.
>
> Previously, we have compared conventional classification approaches in the appendix, including balance sampling and focal loss. Following your valuable suggestion, we have further added the comparisons with Hard-negative sample mining and Online Hard Example Mining (OHEM) in the revision.
> **We have moved these results from the appendix back to the main paper (Table 9) in the revision**. The results are as follows:
>
> | Method                      | FLARE validation      |
> |-----------------------------|----------|
> | Cross-entropy loss          | 37.6     |
> | Balanced cross-entropy loss | 37.8     |
> | Focal loss                  | 36.5     |
> | Hard-Negative Sampling      | 35.0     |
> | OHEM                        | 39.5     |
> | GF-Screen (RL)              | **56.7** |
>
> References:
> - Pedro et al. Object detection with discriminatively trained part-based models. TPAMI, 2009
>
> - Abhinav et al. Training region-based object detectors with online hard example mining. CVPR, 2016
>
> We observe that simply advancing the classification approaches cannot effectively improve the performance. We conclude that conventional classification approaches **still rely on the coarse volume-level classification labels**, while more precise supervision signals are required for training.
>
> **[Adoption of RL]**: To this end, we propose to leverage the segmentation results to provide more precise supervision for the Glance model.
> Finally, given that the selection operation of the Glance model is non-differentiable in the segmentation training, we thereby adopt RL for training the Glance model.
>
> Thanks for your valuable consideration. We have highlighted these discussions and experiments in the revision.

---

> ### Author Response · Authors · 2025-11-20
> **[2/3] Response to Reviewer bZ4d**
>
> **Comment 2**: Using segmentation performance as the reward signal: a high segmentation score may reflect an "easy" sub-volume, ... a low segmentation score might indicate a challenging but clinically critical case ... risks biasing toward easy regions and discarding hard, ambiguous, yet important ones.
>
> **Response 2**: Thanks a lot for your professional insights. **We are delighted to see that our findings exactly align with your perspective, which is also a key point described in our manuscript!**
>
> We would like to highlight that **we did not directly use the segmentation score as rewards**. As discussed in Section 3.3 (Reward design):
>
> *"Our reward is binary and determined by the overlap between the predicted segmentation and the ground-truth lesion mask. Once the segmentation prediction overlaps with the lesion mask, it returns rewards r=1; otherwise, it returns r=0 ...
> We have further explored using the segmentation DSC for a more granular reward signal, but we observed that the performance is worse. We conclude that **detection of the lesion presence is more important in the Glance model**, while DSC varies significantly with lesion complexity (e.g., types, sizes, and positions). Thus, to prioritize detection accuracy in the Glance model, we use the binary detection reward during RL training."*
>
> As you wisely point out, we found that using segmentation scores (DSC) as rewards yield worse results. Thus, we design a lenient detection reward function using the overlap between the segmentation prediction and lesion mask. Notably, **previous cancer screening works (Cao et al., Hu et al.) also employed such detection methods to avoid discarding hard yet important cases that with lower segmentation scores, which substantiates the reasonableness of our reward design.**
>
> References:
> - Cao et al. Large-scale pancreatic cancer detection via non-contrast ct and deep learning. Nature Medicine, 2023
>
> - Hu et al. AI-based large-scale screening of gastric cancer from noncontrast ct imaging. Nature Medicine, 2025
>
> We have conducted ablation studies regarding this part. **We are encouraged that our findings exactly align with your professional perspective**, *i.e.*, directly using segmentation scores as rewards is not appropriate. **We have moved these ablation studies from the appendix back to the main paper** (Table 9), brief results as follows:
>
> | Method                                                 | FLARE validation  |
> |:-------------------------------------------------------| :--- |
> | Group Relative Learning (Segementation DSC as rewards) | 43.2 |
> | Group Relative Learning (Reward function Eq.2)         | 56.7 |
>
> We sincerely appreciate for pointing out our omission to highlight this important part. In addition, we also found that your summary in comment 2 is more clear and straightforward. **To ensure our potential readers can also benefit from your professional insights, we have highlighted your perspective in our manuscript** (also in Section 3.3-Reward design):
>
> *"... We conclude that detection of the lesion presence is more important in the Glance model, while DSC varies significantly with lesion complexity (e.g., types, sizes, and positions). Concretely, a high segmentation DSC may simply reflect an easy sub-volume (e.g., one with clear boundaries or canonical orientation). Conversely, a low segmentation DSC might indicate a challenging but clinically critical case (e.g., partial lesions or suboptimal viewing angles). Thus, to prioritize detection accuracy in the Glance model, we use the binary detection reward during RL training. Previous cancer screening works (Cao et al., Hu et al.) also employed such detection methods to avoid discarding hard yet important cases, substantiating the reasonableness of our reward design."*

---

> ### Author Response · Authors · 2025-11-20
> **[3/3] Response to Reviewer bZ4d**
>
> **Comment 3**: The paper’s layout could be optimized for impact. Table 1 could be moved to the supplementary material for more critical content, particularly ablation studies and a conclusion section...
>
> **Response 3**: Thanks for your thorough review and valuable suggestion, improving the readability of our paper. We have noted that our previous organization has also led to some major concerns of other reviewers. Following your suggestion, we have moved dataset statistics to the appendix. The ablation studies have been moved from the appendix back to the main paper, and we have also added a thorough conclusion section (Section 5) in the revision.
>
> ___
>
> **Comment 4**: The work primarily constitutes an application of the GRPO to a new domain, rather than a significant algorithmic advance.
>
> **Response 4**: Thanks for your valuable comment, making our contributions clearer.
>
> First, we would like to clarify that our main contribution lies in developing the first RL framework for tackling the challenges in pan-cancer screening. The adaptation of GRPO only constitutes the optimization part of the whole framework, which is tailored to the optimization bottleneck in our vision task.
>
> Second, we would like to highlight that the adaptation of GRPO is specifically designed for addressing the challenges of our task, which is not a straightforward application. Specifically:
> - **Previous GRPO works heavily rely on the usage of LLMs for generating a group of candidate responses**, which is not feasible for our vision task.
> - Without requirements of LLMs, GF-Screen leverages the group of sub-volumes for group relative comparisons, effectively shifting the paradigm from NLP to vision tasks.
>
> Following your kind consideration, we have highlighted these parts in the revision.
>
> ___
>
> **Comment 5**: The term "data curation"... Label inconsistencies or annotation discrepancies.
>
> **Response 5**: Thanks for your careful review and for pointing out our imprecise wording, ensuring the rigor of our manuscript.
> We agree that the word "curate" may lead to potential misunderstanding. We would like to clarify that we did not mean to over-claim, but only wanted to demonstrate that we evaluate our method on a large-scale dataset.
>
> Following your valuable suggestion, we have corrected "curate" as "aggregate". **The word "curate" exists only once in our paper (introduction Line 101), we have revised it as follows**:
>
> *"To evaluate the effectiveness, we **aggregate** 5,117 CT scans..."*
>
> Following your valuable suggestion, we have carefully checked the manuscript to avoid imprecise words and phrases.
>
> [Label inconsistencies or annotation discrepancies]: For the pan-cancer screening task, we aim to develop one universal model for all types of lesions. Thus, for different source datasets, we assign the labels of all lesion types as 1 and the background as 0, following the settings of the FLARE challenge and previous works for fair comparisons.
>
> ___
>
> **Comment 6**: Was this comparison conducted under equal conditions? (the same training data? the use of external data?)
>
> **Response 6**: Thanks for pointing out our omission to provide this important detail. **Yes, the comparison is conducted under equal conditions**. The official challenge holds a list of datasets for participants to conduct training. No external data is allowed for training. We have highlighted this information in the revision (Section 6.1 Dataset descriptions).

---

### Official Review · Reviewer_xJb3 · 2025-11-01

**Soundness:** 3
**Presentation:** 3
**Contribution:** 3
**Rating:** 6
**Confidence:** 3

**Summary:**

This paper proposes GF-Screen, a novel reinforcement learning (RL) framework designed for pan-cancer screening from large-scale CT scans. The method mimics the radiologist’s diagnostic process — “glance” to locate suspicious regions and “focus” to perform detailed lesion segmentation. GF-Screen integrates a Glance model for sub-volume selection and a Focus model for segmentation. The Glance model is optimised through a Group Relative Learning (GRL) paradigm, where rewards are derived from segmentation outcomes of the Focus model. Experiments on 5,117 CT scans from 23 public datasets, covering nine lesion types, show substantial improvements over previous approaches, achieving the top rank on the MICCAI FLARE25 challenge, with reported +25.6% DSC and +28.2% NSD compared to the FLARE24 champion solution. The paper emphasises efficiency gains (5.7× faster inference) and reduced false positives through selective attention to diseased regions.

**Strengths:**

Strengths
1. The “glance and focus” paradigm is a clever conceptual and algorithmic adaptation of clinical diagnostic reasoning to AI-based screening.
2. The paper provides a well-defined pipeline linking reinforcement learning and segmentation, with precise mathematical formulations for the GRL optimisation objective.
3. Addressing pan-cancer screening rather than single-organ segmentation is highly ambitious and clinically meaningful.
4. The study presents extensive experiments on both internal and external datasets, including strong validation on the FLARE25 leaderboard.
5. The demonstrated 5.7× reduction in computational cost without accuracy loss is a notable contribution for real-world clinical deployment.
6. The quantitative results convincingly show that GF-Screen outperforms a range of established baselines, including nnUNet, SwinUNETR, VoCo, and PASTA.
7. The adaptation of GRL to vision-based medical tasks is novel and extends recent RL developments in a meaningful way.

**Weaknesses:**

Weaknesses
1. While the proposed framework is well-engineered, its novelty may primarily lie in combining existing ideas (sub-volume selection, segmentation, RL reward optimisation) rather than introducing a fundamentally new theoretical concept.
2. The reinforcement learning formulation, especially the reward design and advantage estimation, is largely empirical. There is little discussion of convergence, stability, or theoretical motivation for using group relative learning in medical imaging.
3. The paper would benefit from more detailed ablation studies isolating the effects of the RL component, the GRL mechanism, and the segmentation backbone. Interpretability analyses (e.g., qualitative localisation or attention visualisations) are limited.
4. While the authors state that code will be released, reproducibility remains uncertain. Essential hyperparameters (e.g., coefficients α, β, learning rates, or update frequency of G_ref) should be clearly stated in the main paper rather than deferred to the appendix.
5. The dataset curation process aggregates 23 datasets with differing acquisition protocols and lesion distributions. Without normalisation or harmonisation details, there is a risk that model performance may rely on dataset-specific biases.
6. The training process of the Glance model may be unstable due to non-differentiable sub-volume selection, but the paper does not report training curves or sensitivity analyses beyond one brief figure.
7. Despite strong quantitative results, the absence of clinician-in-the-loop evaluation or human–AI comparison weakens the practical validation of the proposed clinical analogy.
8. The paper is technically dense, with long paragraphs and numerous references, which may hinder readability for non-specialist reviewers. Some restructuring and pruning would improve clarity.

**Questions:**

plz see my detaild comments above

---

> ### Author Response · Authors · 2025-11-20
> **[1/3] Response to Reviewer xJb3**
>
> We sincerely appreciate your thorough review and constructive suggestions for improving our work. We apologize for placing some important settings, ablation studies, and visualization results in the appendix previously, which may have caused your reasonable concerns. We have reorganized the paper and highlighted these important parts in the revision. Our point-by-point responses are below.
>
> ___
>
> **Comment 1**: Combining existing sub-volume selection, segmentation, and reward optimisation, instead of a new theoretical concept.
>
> **Response 1**: Thanks for your valuable comment, making our contributions clearer. First, we would like to highlight that the core novelty of GF-Screen lies not in the combination of these components, but in **the whole RL framework driven by the specific challenges in pan-cancer screening**. We synergetically integrate these basic components to establish our framework, which is not a straightforward combination.
>
> In this work, we did not aim to propose a new theory for RL. Our contribution is introducing the first RL framework specifically designed for pan-cancer screening, which we believe is **a novel and practical conceptual breakthrough in this field**. For your valuable consideration, we have highlighted this part in the revision, as follows (Section 5 Conclusion):
>
> *"In summary, our work did not aim to propose a new theory for RL. Instead, our contribution is introducing the first RL framework specifically designed for pan-cancer screening, which is a novel and practical conceptual breakthrough in this field."*
>
> ___
>
> **Comment 2**: Discussion of convergence, stability, or theoretical motivation for using GRL in medical imaging.
>
> **Response 2**: Thanks a lot for your valuable concerns and professional insights, making our discussions more comprehensive. First, we would like to discuss the motivation of GRL in our pan-cancer screening task, which is highlighted in Sections 3.2 and 3.3:
>
> - **[Eliminating the training of value models in RL]**: Traditional RL algorithms like PPO require a large value model for advantage estimation, which is difficult to train.
> - **[The sub-volumes group naturally enables group comparisons]**: To eliminate value models, previous GRPO methods heavily rely on Large Language Models to generate a group of candidate answers for advantage estimation. In our GF-Screen, the sub-volume group for selection naturally enables us to conduct group relative advantage comparison, which is the key motivation to our GRL.
>
> Then, to evaluate the stability, we have conducted five times of experiments (Tables A6 and A7) and analyzed the t-test p-values (Table A8). We have also analyzed the training convergence by the loss curves of GRL (Figure A6). Our evaluation results on a large-scale validation dataset show that the performance of GRL is stable.
>
> Thanks again for your valuable comment. We have highlighted all these parts in the revision.

---

> ### Author Response · Authors · 2025-11-20
> **[2/3] Response to Reviewer xJb3**
>
> **Comment 3**: The paper would benefit from more detailed ablation studies isolating the RL component, the GRL mechanism, and the segmentation backbone.
>
> **Response 3**: Thanks a lot for your valuable suggestion! **We placed these ablation studies in the appendix previously, and we have moved them back to the main paper following your suggestion**. Specifically, our ablation studies include:
>
> - Ablation studies of the PPO objective (Move to Table 9)
> - Ablation studies of reward functions and objectives in GRL (Move to Table 9)
> - Ablation studies of efficiency (Move to Table 10)
> - Ablation studies of segmentation backbone (Move to Table 11)
> - Ablation studies of the Glance model backbone (Move to Table 12)
> - Ablation studies of the number of sub-volumes *N* in group comparison (Move to Table 12)
>
> For your valuable consideration, we have re-organized our manuscript and highlighted these parts in the main paper. Thanks again for your suggestion!
>
> ___
>
> **Comment 4**: Interpretability analyses (e.g., qualitative localisation or attention visualisations) are limited.
>
> **Response 4**: **We apologize that we placed the visualization analysis in the appendix, since these figures occupy a lot of space**. Specifically,
>
> - Qualitative localisation visualisations (Appendix Fig.A2)
> - Effectiveness of reducing false positives (Appendix Fig.A3)
> - Effectiveness of selecting optimal views (Appendix Fig.A4)
> - Segmentation results (Appendix Fig.A5)
> - Loss curves of GRL (Appendix Fig.A6)
>
> For your valuable consideration, we have highlighted these figures in the main paper. Thanks again for your valuable suggestion!
>
> ___
>
> **Comment 5**: Code reproducibility remains uncertain. Essential hyperparameters should be clearly stated in the main paper.
>
> **Response 5**: Thanks for your valuable suggestion! **We promise that our codes will be released for reproducibility**.
> Following your valuable suggestion, **we have moved the implementation and essential hyperparameters settings from the appendix to the main paper** (Section 4.1, including α, β, learning rates, update frequency of G_ref), ensuring followers can easily reproduce our work.
>
> ___
>
> **Comment 6**: Normalisation or harmonisation details of datasets. Model performance may rely on dataset-specific biases.
>
> **Response 6**: Thanks for your valuable concern, improving the rigor of our manuscript. **We have moved the dataset pre-processing details from the appendix to the main paper (Section 4.1).**
>
> **[Dataset-specific biases]**: First, we would like to point out that GF-Screen consistently outperforms previous methods on 23 datasets with different acquisition protocols, substantiating that our method does not rely on dataset-specific biases.
> In addition, we would like to highlight that, **there are 7 external datasets in our evaluation, which are unseen in training**. Extensive experiments have demonstrated that our method can achieve superior performance on unseen external datasets (including the public leaderboard results), indicating the effectiveness of GF-Screen does not rely on dataset-specific biases.
>
> For your kind concern, we have highlighted this issue in the revision (Section 4.1):
>
> *"To demonstrate that the effectiveness of GF-Screen does not rely on dataset-specific biases, we involve 7 external datasets in evaluation, which are unseen in training."*

---

> ### Author Response · Authors · 2025-11-20
> **[3/3] Response to Reviewer xJb3**
>
> **Comment 7**: The training progress may be unstable. Report training curves or sensitivity analyses beyond one brief figure.
>
> **Response 7**: Thanks a lot for your valuable suggestion, ensuring the rigor of our experiments. To evaluate the stability of training, we have reported the results of five experimental runs in Appendix Table A6 and A7 as follows (segmentation DSC and detection F1-score), showcasing that the training of GF-Screen is stable. We have also conducted statistical analysis of our results by reporting the t-test p-values compared with the best-competing method (Table A8).
>
> For your valuable concern, we have added more sensitivity analyses of training in Appendix Figure A6. Specifically, we randomly split our training dataset into five subsets and perform training to analyse the loss curves of RL. We observe that the resulting curves closely align with those in prior GRPO literature, demonstrating the robustness of our training process.
>
> Thanks again for your valuable suggestion! We have highlighted all these parts in the revision.
>
> | Method    | Lung tumor | Lung nodule | Pleural effusion | COVID-19 infection | Liver tumor | Pancreas tumor | Kidney tumor | Adreno. carcinoma | Colon tumor |
> |-----------|------------|-------------|------------------|-------------------|-------------|---------------|--------------|------------------|-------------|
> | GF-Screen | 57.5±0.8   | 55.4±0.3    | 45.0±0.3         | 69.0±0.7          | 67.6±0.5    | 49.2±0.6      | 75.3±0.3     | 91.5±1.1         | 38.1±1.2    |
>
> | Method    | Lung tumor | Lung nodule | Pleural effusion | COVID-19 infection | Liver tumor | Pancreas tumor | Kidney tumor | Adreno. carcinoma | Colon tumor |
> |-----------|------------|-------------|------------------|--------------------|-------------|----------------|--------------|-------------------|-------------|
> | GF-Screen | 96.4±0.2   | 91.9±0.1    | 96.6±0.1         | 100.0±0.0          | 98.2±0.2    | 95.1±0.2       | 100.0±0.0    | 100.0±0.0         | 85.0±0.5    |
>
> | P-values       | Lung tumor | Lung nodule | Pleural effusion | COVID-19 infection | Liver tumor | Pancreas tumor | Kidney tumor | Adreno. carcinoma | Colon tumor |
> |----------------|------------|-------------|------------------|--------------------|-------------|----------------|--------------|------------------|-------------|
> | Segmentation DSC | 4.54×10⁻³ | 3.01×10⁻⁴ | 7.46×10⁻⁵ | 3.34×10⁻³ | 8.99×10⁻³ | 3.35×10⁻⁴ | 7.76×10⁻³ | 9.01×10⁻⁴ | 4.89×10⁻³ |
> | Detection F1-scores| 1.80×10⁻³ | 2.06×10⁻⁵ | 4.11×10⁻³ | 2.54×10⁻² | 1.53×10⁻² | 9.13×10⁻³ | 8.10×10⁻² | 1.15×10⁻² | 6.80×10⁻⁴ |
>
> ___
>
> **Comment 8**: Absence of clinician-in-the-loop evaluation or human–AI comparison.
>
> **Response 8**: We sincerely appreciate your profound suggestion. We fully acknowledge the importance of clinical validation and human-AI comparison. However, such a study requires both significant time from expert radiologists and formal ethical approval for using clinical data, which was not feasible within the timeline. We have therefore explicitly listed this as a future plan in the revision (Section 6.10 Future Directions):
>
> *"In the future, we will also include clinician-in-the-loop evaluation and human–AI comparison to strengthen the evaluation of our method. We will work with our collaborating hospitals, *e.g.*, conduct a reader study using real-world data. We will compare the diagnostic efficiency and accuracy of radiologists with and without AI assistance from GF-Screen."*
>
> ___
>
> **Comment 9**: The paper is technically dense. Some restructuring and pruning would improve clarity.
>
> **Response 9**: Thank you very much for your careful review and valuable suggestion! We have carefully revised the manuscript by removing redundant phrases and sentences, improving the clarity and readability.

---

### Official Review · Reviewer_HUYR · 2025-11-01

**Soundness:** 3
**Presentation:** 3
**Contribution:** 3
**Rating:** 4
**Confidence:** 3

**Summary:**

This paper presents GF-Screen, a novel framework for pan-cancer screening in large-scale CT volumes. The core problem addressed is the significant challenge of locating small, diverse lesions within vast, mostly healthy, anatomical regions—an issue of extreme foreground-background imbalance. The proposed solution is inspired by the "glance and focus" strategy of radiologists. It consists of two main components: a Glance model that acts as a selector to identify promising sub-volumes, and a Focus model that performs detailed segmentation only on the selected regions. The key technical contribution is the method for training the Glance model. Since the selection action is non-differentiable, the authors employ a RL approach where the Focus model's segmentation output provides a reward signal to the Glance model. A novel GRL paradigm is introduced to optimize this policy. The method is validated on a large, diverse dataset aggregated from 23 public sources and shows outstanding results, notably leading the MICCAI FLARE25 challenge validation leaderboard by a significant margin and demonstrating a 5.7x reduction in computation.

**Strengths:**

1. The results are the most significant strength. Leading the FLARE25 challenge validation leaderboard and outperforming the FLARE24 champion solution by such a large margin (+25.6% DSC) is a truly remarkable achievement.
2. The method directly addresses a critical bottleneck in deploying AI screening models: computational cost and false positives. By intelligently discarding healthy regions, the 5.7x computational saving is a major practical breakthrough, making large-scale screening far more feasible.
3. The use of RL to couple a detection/localization network (Glance) with a segmentation network (Focus) is elegant. The segmentation result (a binary overlap reward ) is used as a rich, task-specific feedback signal for the localization model, which is more sophisticated than training the localizer on simple binary "lesion present/absent" labels. The proposed Group Relative Learning (GRL) appears effective, as shown in the ablations.

**Weaknesses:**

1. The paper introduces GRL as a "novel" paradigm, but its precise mechanism and distinction from existing policy gradient methods (like PPO, which is cited) could be clearer. The loss function in Eq. 3 looks like a PPO-style clipped objective. The novelty seems to be in the advantage estimation or the "group relative comparison", but this specific aspect is not explained in sufficient detail. This lack of clarity is particularly problematic concerning the core RL mechanism. For instance, the paper needs to explicitly detail the action space. The Glance model's role is to select (1) or discard (0) a sub-volume, which is a discrete, non-differentiable step. The paper should clarify how this binary decision is derived from the policy network. Is the model's output 'o' (as referenced around line 206) a probability (e.g., a sigmoid output) from which an action is sampled? If 'o' is treated as a discrete 0/1 output directly, it's unclear how the policy gradient can be backpropagated for optimization. A more detailed explanation is required of what exactly constitutes the 'action' in the RL framework (e.g., the binary selection), how this action is stochastically sampled from the policy during training to enable exploration, and how the underlying policy (the probability distribution over actions) is updated via the GRL objective.
2. The primary justification for the RL approach is the failure of a standard classification (CE) baseline . However, the paper does not specify how this baseline was trained. Given the stated problem of "severe foreground-background imbalance", a simple CE loss is expected to fail. A fair comparison would require comparing GRL to a strong classification baseline

**Questions:**

See weakness 1.

---

> ### Author Response · Authors · 2025-11-20
> **[1/2] Response to Reviewer HUYR**
>
> We are highly grateful for your professional insights and valuable suggestion, which have significantly enhanced the clarity of our method. We have carefully revised our manuscript following your suggestions, especially strengthening the methodology details. Our point-to-point responses are as below.
>
> ___
>
> **Comment 1:** The precise mechanism of GRL and its distinction from existing policy gradient methods (like PPO) could be clearer.
>
> **Response 1:** Thanks for your constructive suggestion, improving the clarity of our method. We have carefully revised our discussions in Section 3.3. First, we would like to clarify that our GRL is an adaptation of GRPO (Shao et al.) from NLP to vision tasks, which is distinct from PPO. Specifically, the differences are as follows:
>
> - **[PPO]**: PPO requires training a value model as a critic to estimate the advantages, which brings a substantial memory and computational burden.
>
> - **[GRPO]**: GRPO (Shao et al., 2024) foregoes the value model, instead using Large Language Models (LLMs) to generate a group of candidate answers and estimate the relative advantages, significantly reducing training resources. And yes, as you point out, GRPO still uses a PPO-style clipped objective.
>
> - **[GRL]**: Without the use of LLMs, we adapt the paradigm of GRPO from NLP tasks to address the specific challenges in pan-cancer screening. In GF-Screen, the group of sub-volumes naturally provides comparable candidates for us to conduct group relative comparison like GRPO, which enables us to eliminate the requirement of value models like PPO. This motivates us to develop the GRL paradigm (Section 3.3 and Figure 5).
>
> Reference: Shao et al. Deepseekmath: Pushing the limits of mathematical reasoning in open language models. 2024.
>
> Concretely, in GRL, we first compute the rewards for each sub-volume (Eq.2) and calculate the group relative advantages (Eq.3) within the sub-volume group, eliminating the requirement of value models to estimate advantages. Finally, we optimize the PPO-like objective (Eq.4). The details are described in Section 3.3 and Figure 5. In response to your valuable consideration, we have carefully revised Section 3.3 to improve the clarity of our method.
>
> We have also compared our method with the PPO algorithm in the experiments. For your valuable concern, we have moved this part from the appendix to the main paper (Table 9).
>
> ___
>
> **Comment 2:** The lack of clarity: Need to explicitly detail the action space...
>
> **Response 2:** Thanks a lot for your careful review and for pointing out our omission to provide this important detail, ensuring the rigor of our manuscript.
>
> **The final layer of our policy model (Glance model) is a 2-unit layer with softmax activation**, outputting a probability distribution over the two actions (select (1) or discard (0)). During training, the output *'o'* is stochastically sampled from this softmax distribution. As with standard RL algorithms, the GRL objective directly optimizes the underlying probability distribution using the policy gradient, bypassing the non-differentiability of the sampling step.
>
> We sincerely appreciate you reminding us to describe this important step, ensuring the clarity of our method. **We have highlighted this important detail in the revision (Section 3.3).**

---

> ### Author Response · Authors · 2025-11-20
> **[2/2] Response to Reviewer HUYR**
>
> **Comment 3:** How the CE baseline was trained?
>
> **Response 3:** Thanks for your comment. The details of the CE baseline were described at the beginning of Section 3.2 and Eq. 1. To avoid being overlooked by our potential readers, we have highlighted this part in the revision. For your concern, we briefly introduce it here:
>
> *"Since we have lesion masks for supervision, a straightforward way is to degrade the lesion masks m as binary categories y (with lesions or without lesions) for each sub-volume v_i, then employing a typical cross-entropy loss for classification training as follows:*
>
> \begin{equation}
> \mathrm{CE}(o_{i},y_{i}) = -y_{i} \log(o_{i}) - (1-y_{i})(1-\log(o_{i})), \quad o_{i} = G(v_{i}),
> \end{equation}
>
> *where o_i denotes the selection outputs of the Glance model G ..."*
>
> ___
>
> **Comment 4:** A simple CE loss is expected to fail, require comparing a strong classification baseline.
>
> **Response 4:** Thanks a lot for your valuable suggestion! We have compared balanced CE loss and Focal loss in the experiments previously. Following your suggestion, we have further added the comparisons with Hard-Negative Sampling and OHEM in the revision.
> We sincerely apologize for placing these important results in the appendix previously, which may have led to your reasonable concern. **We have moved these results from the appendix back to the main paper (Table 9) in the revision**.
>
> The results are as follows: we observe that simply advancing the classification baseline cannot effectively improve the performance.
> We conclude that stronger classification baselines still rely on the coarse volume-level classification labels.
> These results motivate us to introduce more precise supervision for training. Thus, we propose to leverage the fine-grained segmentation result as rewards for RL training, which can provide more precise supervision for the Glance model.
>
> | Method                      | FLARE validation      |
> |-----------------------------|----------|
> | Cross-entropy loss          | 37.6     |
> | Balanced cross-entropy loss | 37.8     |
> | Focal loss                  | 36.5     |
> | Hard-Negative Sampling      | 35.0     |
> | OHEM                        | 39.5     |
> | GF-Screen (RL)              | **56.7** |
>
> Following your valuable suggestion, we have highlighted these results in the revision. Thanks again for your constructive suggestions to improve our work!

---

### Author Response · Authors · 2025-11-20
**Response to AC and Reviewers**

We sincerely appreciate the AC and reviewers for your dedicated efforts in reviewing, which have significantly improved the quality of our manuscript. **We are genuinely encouraged by the generous commendations and positive feedback from the reviewers**:

- Reviewer HUYR: *"a novel framework, elegant and effective, truly remarkable achievement, addresses a critical bottleneck, practical breakthrough"*.
- Reviewer xJb3: *"novel and meaningful way, clever conceptual and algorithmic adaptation, well-defined pipeline, clinically meaningful, strong validation, convincing quantitative results, notable contribution for real-world clinical deployment"*.
- Reviewer bZ4d: *"practical step in accurate, scalable, and efficient pan-cancer screening, easy to follow, extensive evaluation"*.
- Reviewer Cyrm: *"innovative and well-motivated idea, comprehensive evaluation, superior performance"*.

We apologize that we overlooked highlighting some important parts in our main paper and appendix previously, which may have led to some reasonable concerns of our reviewers. Following your valuable comments, we have made thorough revisions to our manuscript. We are eager to engage with your professional insights in the rebuttal, which we believe represents an invaluable opportunity for academic exchange.

---

> ### Author Response · Authors · 2025-11-27
> **A Gentle Reminder for Reviewing**
>
> Dear AC and Reviewers,
>
> Happy Thanksgiving! Once again, we sincerely thank you for your thorough review and the valuable suggestions you provided to improve our work.
>
> As the rebuttal deadline is approaching, we would be immensely grateful if you could take a moment to review our responses and the revised manuscript. Your feedback on whether any concerns remain unresolved would be greatly appreciated.
>
> Thanks for your time and support!

---

### Author Response · Authors · 2025-12-03
**Summary of Rebuttal**

Dear AC and Reviewers,

It is a great pity that we were unable to receive your feedback during the rebuttal period. No matter what the final result is, we sincerely appreciate your generous commendations and constructive suggestions, which have significantly improved our manuscript. **We are genuinely encouraged by the reviewers' heartening recognition of our work's novelty and contributions:**

- [Reviewer HUYR]: ***"a novel framework, elegant and effective, truly remarkable achievement, addresses a critical bottleneck, practical breakthrough"***.
- [Reviewer xJb3]: ***"novel and meaningful way, clever conceptual and algorithmic adaptation, well-defined pipeline, clinically meaningful, strong validation, convincing quantitative results, notable contribution for real-world clinical deployment"***.
- [Reviewer bZ4d]: ***"practical step in accurate, scalable, and efficient pan-cancer screening, easy to follow, extensive evaluation"***.
- [Reviewer Cyrm]: ***"innovative and well-motivated idea, comprehensive evaluation, superior performance"***.

Here, we summarize our responses as follows:

- [Reviewer HUYR]:
  - Comment 1: We clarified the distinction from previous policy gradient methods like PPO.
  - Comment 2: We added the details of the action space.
  - Comment 3: We highlighted the details of the CE baseline in Section 3.2.
  - Comment 4: We moved the comparisons with classification approaches from the appendix to the main paper.
- [Reviewer xJb3]:
  - Comment 1: We clarified the contribution of our work (the first RL framework designed for pan-cancer screening).
  - Comment 2: We highlighted the motivation of GRL.
  - Comment 3: We moved the ablation studies from the appendix to the main paper.
  - Comment 4: We highlighted the visualization analysis in the appendix.
  - Comment 5: We moved the implementation and essential hyperparameters settings from the appendix to the main paper.
  - Comment 6: We moved the dataset pre-processing details from the appendix to the main paper.
  - Comment 7: We added the sensitive analysis to the paper.
  - Comment 8: We highlighted the importance of clinical evaluation.
  - Comment 9: We carefully restructured and pruned the manuscript.
- [Reviewer bZ4d]:
  - Comment 1: We clarified the motivation of using RL and moved the related ablation studies from the appendix to the main paper.
  - Comment 2: We clarified the misunderstanding of our reward function and highlighted the related discussion and ablation studies.
  - Comment 3: We moved the ablation studies from the appendix to the main paper.
  - Comment 4: We clarified the contribution of our work (the first RL framework designed for pan-cancer screening).
  - Comment 5: We revised the word "curation".
  - Comment 6: We added the details of challenge setting.
- [Reviewer Cyrm]:
  - Comment 1: We added the supplementary details of Equation 4.
  - Comment 2: We clarified the details of evaluation datasets.
  - Comment 3: We added the protocols and device of the used datasets.
  - Comment 4: We added the ablation studies of sliding window sizes.
  - Comment 5: We added the statistical analysis of the results.

**In summary, we are confident that our responses have adequately addressed the concerns of reviewers.** Thank you again for your valuable time in reviewing!

---

### Meta-Review · Area_Chair_HSqd · 2025-12-27

**Summary:**

The paper adapts Glance and Focus approach (see e.g. Wang et al 2020) to medical image classification by making significant practical improvements such as using Group Relative Learning.

Reviewers highlighted the strong empirical results on the MICCAI FLARE25 validation leaderboard. Reviewers also appreciated the improvement in speed (~5x) thanks to the discarding mechanism (though similar in spirit to GMIC global vs local network).

Reviewers were concerned by novelty, which I think could be addressed by more clearly describing prior work such as Wang et al.

Reviewers were also concerned that RL could be a suboptimal method. The motivation for using RL in LLMs is quite different. Here, one could imagine training the method using supervised signals. Authors have made some progress towards addressing this issue by comparing to Hard Negative Sampling, but more methods are available (e.g. GMIC).

Regretfully, Reviewers have not engaged during the discussion phase. As such, I have made determination whether Authors have properly addressed the comments.

All in all, this is a solid though borderline application-focused paper that clears the bar for acceptance. I am happy to recommend accepting the work.

**Reviewer Concerns:**

A major concern raised by Reviewer bZ4d and Reviewer HUYR was not comparing to enough baselines. Authors have added comparison to Hard Negative Sampling (which underperforms their method) but the paper would benefit from comparing to more non-RL methods (e.g. GMIC).

Reviewer bZ4d had a very specific concern that using segmentation quality (DSC) as a reward would bias the model toward "easy" shapes and ignore "hard" but critical lesions. The authors effectively rebutted this by showing that binary detection rewards worked better than DSC rewards (53.1% vs 43.2%).

Reviewer xJb3 and Reviewer bZ4dz raised concerns regarding novelty. Indeed the paper is mostly an application of existing concepts, and the overall idea of “Glance and Focus” has been explored before (e.g. Wang et al, https://arxiv.org/abs/2010.05300). The paper should be a bit clearer about the relation to previous works, which should be more clearly discussed in the Introduction section.

**Reviewer Scores:**

Reviewers gave scores:

- **Reviewer HUYR:** 4 (Marginally below acceptance)
- **Reviewer xJb3:** 6 (Marginally above acceptance)
- **Reviewer bZ4d:** 4 (Marginally below acceptance)
- **Reviewer Cyrm:** 10 (Strong accept)

I believe some of the reviewers might have increased the score had they engaged in the discussion.

---

### Decision · Program_Chairs · 2026-01-26

Accept (Poster)